# Unsupervised Representation Learning for 3D Mesh Parameterization with Semantic and Visibility Objectives

**AmirHossein Zamani**[1,2,3] *     **Bruno Roy** [1]     **Arianna Rampini** [1]

[1] Autodesk Research     [2] Mila – Quebec AI Institute     [3] Concordia University

Project page: https://ahhhz975.github.io/Automatic3DMeshParameterization/

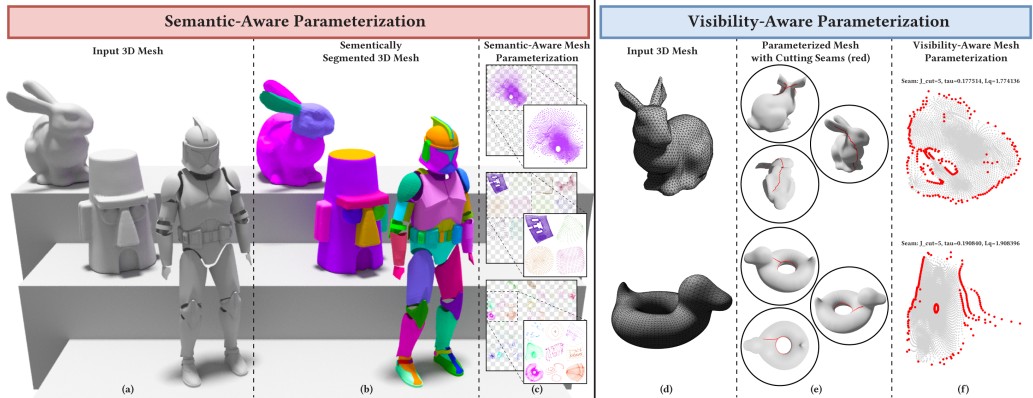

Figure 1: 3D mesh parameterizations generated by our semantic-aware (left) and visibility-aware (right) pipelines. **Semantic-Aware (left, Sec. 3.3):** To encourage semantically coherent UV charts for easier texture editing, given an input mesh (a), we design a *partition-and-parameterize* strategy, (b) compute a per-vertex semantic partition, (c) learn a geometry-preserving UV parameterization (Sec. 3.2) independently for each part to obtain UV islands, and finally aggregate and pack these islands into a unified UV atlas (insets). **Visibility-Aware (right, Sec. 3.4):** To encourage seamless UV mappings, the pipeline (d) takes an input 3D mesh, jointly (e) guides cutting-seam placement (red curves), extracts corresponding UV boundary points (red dots), and (f) estimates a global geometry-preserving parameterization. This steers cutting seams toward less-visible (more occluded) surface regions, producing more visually seamless UV maps.

## Abstract

Recent 3D generative models produce high-quality textures for 3D mesh objects. However, they commonly rely on the heavy assumption that input 3D meshes are accompanied by manual mesh parameterization (UV mapping), a manual task that requires both technical precision and artistic judgment. Industry surveys show that this process often accounts for a significant share of asset creation, creating a major bottleneck for 3D content creators. Moreover, existing automatic methods often ignore two perceptually important criteria: (1) semantic awareness (UV charts should align semantically similar 3D parts across shapes) and (2) visibility awareness (cutting seams should lie in regions unlikely to be seen). To overcome these shortcomings and to automate the mesh parameterization process, we present an unsupervised differentiable framework that augments standard geometry-preserving UV learning with semantic- and visibility-aware objectives. For semantic-awareness, our pipeline (i) segments the mesh into semantic 3D parts, (ii) applies an unsupervised learned per-part UV-parameterization backbone, and (iii) aggregates per-part charts into a unified UV atlas. For visibility-awareness, we use ambient occlusion (AO) as an exposure proxy and back-propagate a soft differentiable AO-weighted seam objective to steer cutting seams toward occluded regions. By conducting qualitative and quantitative evaluations against state-of-the-art methods, we show that the proposed method produces UV atlases that better support texture generation and reduce perceptible seam artifacts compared to recent baselines. Our implementation code is publicly available at: https://github.com/AHHHZ975/Semantic-Visibility-UV-Param.

*Correspondence to: amirhossein.zamani@mila.quebec

## 1 INTRODUCTION

Texture mapping, also called UV/Mesh parameterization, converts a 3D mesh surface into a 2D image and is a core step in digital 3D content production, enabling texture synthesis, painting, transfer, and high-quality rendering. While recent 3D generative methods produce high-quality textures for 3D mesh objects, they commonly rely on the heavy assumption that input 3D meshes are accompanied by the mesh parameterization process that requires technical precision and artistic judgment. This assumption limits usability for content creators and designers. Moreover, industry surveys show that this process often accounts for a significant share of asset creation, creating a major bottleneck for 3D content creators. To address these issues, numerous methods have been proposed that mainly focus on geometric properties such as bijectivity (Tutte, 1963; Sheffer et al., 2007a), conformality (angle-preservation) (Lévy et al., 2023), stretch (length-preservation) (Sander et al., 2002), and equiareality (area-preservation) (Zou et al., 2011). While these properties are necessary to achieve a high-quality parameterization, they are not sufficient for many downstream applications in content creation and texture synthesis. In particular, two perceptual criteria are often neglected: (1) *semantic awareness*, 2D UV charts should align with semantically meaningful surface 3D parts so that textures designed for a semantic region remain coherent and transferable across 3D shapes; and (2) *visibility awareness*, seams should be placed where they are unlikely to be observed under typical viewpoints and lighting, so that seam artifacts are less perceptible after texturing and rendering. To the best of our knowledge, only limited work has explored these properties individually: semantic-aware UV mapping (Vermandere et al., 2024; Mukherjee et al., 2024) and visibility-aware UV learning (Weill-Duflos et al., 2023; Sheffer & Hart, 2002; Ray et al., 2010). However, none of the existing approaches address both objectives simultaneously. To overcome these shortcomings, we present an unsupervised differentiable framework that augments standard geometry-preserving UV learning with semantic- and visibility-aware objectives. For semantic-awareness, our pipeline (i) segments the mesh into semantic 3D parts, (ii) applies an unsupervised learned per-part UV-parameterization backbone, and (iii) aggregates per-part charts into a unified UV atlas. For visibility-awareness, we use ambient occlusion (AO) (Zhukov et al., 1998) as an exposure proxy and back-propagate a soft differentiable AO-weighted seam objective designed to steer cutting seams toward occluded regions. The proposed pipeline has two stages: (i) a neural surface-parameterization backbone enforces geometry-preserving properties, (ii) semantic-aware and visibility-aware modules provide task-specific (e.g., texture painting) guidance. The main contributions of this study are: (i) We propose an unsupervised, differentiable framework that jointly estimates geometry-preserving UV parameterizations while optimizing semantic- and visibility-aware objectives. (ii) We introduce two novel perceptual objectives that directly connect UV parameterization to texture synthesis: a semantic-aware objective that aligns 2D UV charts with semantically meaningful 3D surface parts, and a visibility-aware objective that places seams in less-visible regions to minimize seam artifacts.

## 2 RELATED WORK

**Traditional UV Parameterization.** These methods aim to flatten 3D surfaces to 2D while minimizing distortion and avoiding overlaps. Early conformal (angle-preserving) techniques such as LSCM (Lévy et al., 2023) and ABF++ (Sheffer et al., 2005) reduce angular distortion via global optimization or iterative angle adjustment. Many approaches also enforce bijectivity and bound distortion: bounded-distortion parameterization (Sorkine et al., 2002) limits local stretch by cutting into patches, bijective free-boundary methods (Smith & Schaefer, 2015) avoid overlaps without fixing boundaries, and SLIM (Rabinovich et al., 2017) scales these guarantees to large meshes. Optimizing cuts and chart segmentation also have been addressed: AutoCuts (Poranne et al., 2017) and OptCuts (Li et al., 2018) jointly optimize seams and UV parameterization to trade off distortion and chart count, while Seamster (Sheffer & Hart, 2002) and Seamless (Liu et al., 2017) place seams to minimize visible texture artifacts by aligning them with features or low-visibility regions. Although classical algorithms provide the geometric foundation for UV mapping, they often treat these goals independently and rarely exploit higher-level objectives (e.g., semantic parts or visibility), which can limit final texture quality.

**Learning-Based UV Parameterization.** The existing methods for UV parameterization can be grouped by input representation: explicit (meshes and point clouds) (Zhao et al., 2025; Zhang et al., 2024b; Srinivasan et al., 2024; Chen et al., 2022; Teimury et al., 2020; Oechsle et al., 2019; Knodt et al., 2023), implicit (NeRFs) (Xu et al., 2024a; Xiang et al., 2021; Srinivasan et al., 2024; Guan et al., 2023; Das et al., 2022), and 3D-Gaussian-splatting-based methods (Xu et al., 2024b). In this work, we focus on explicit (mesh) parameterization, which is widely used in industry. Mesh-

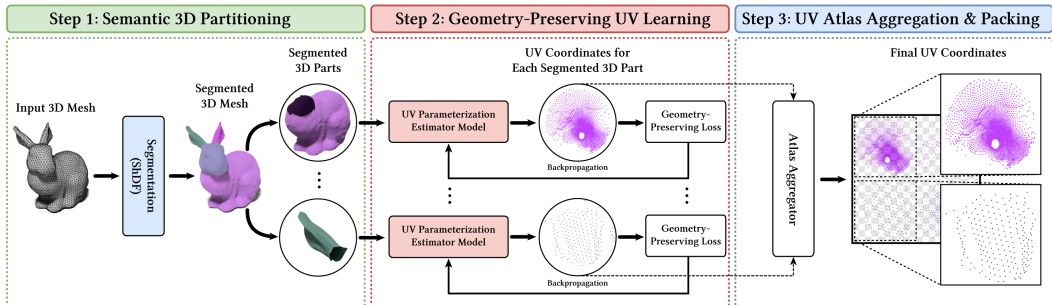

Figure 2: An overview of the training of the proposed semantic-aware UV parameterization method (Sec. 3.3), consisting of three stages: (i) *semantic 3D partitioning*, computes a per-vertex semantic partition of the input mesh using shape diameter function (Appendix A.1); (ii) *geometry-preserving UV learning*, applies the base UV-parameterization backbone (Sec. 3.2) independently to each semantic part to obtain per-part UV islands; and (iii) *UV atlas aggregation and packing*, aggregates and packs these islands into a unified UV atlas.

based learning approaches (Zhao et al., 2025; Srinivasan et al., 2024; Zhang et al., 2024a;b) usually train multi-layer perceptrons (MLPs) to learn UV mappings by optimizing three core criteria: (i) minimizing chart count, (ii) reducing distortion (preserving lengths, angles, and areas), and (iii) enforcing bijectivity via round-trip consistency. These MLPs are trained with carefully designed losses to satisfy those constraints. However, current learning-based methods often *(i) lack semantic awareness*, meaning that cutting seams are chosen only to reduce geometric distortion rather than to align with meaningful parts, and *(ii) ignore visibility when placing seams*, failing to steer cuts toward low-visibility regions and thus producing noticeable texture artifacts.

## 3 METHODOLOGY

We propose a two-stage, unsupervised framework to automatically learn 3D mesh parameterizations with semantic and visibility objectives. The training consists of two stages. In stage (i), *geometry-preservation mesh-parameterization learning* (Sec. 3.2), an MLP-based network is trained with fully differentiable geometric objectives defined in the UV domain to produce an initial UV mapping that satisfies low-distortion requirements. In stage (ii), *learning perceptual objectives* (Sec. 3.3 and Sec. 3.4), we introduce two perceptual objectives: *semantic awareness* and *visibility awareness*. By semantic awareness, we mean that the UV charts produced by stage (i) should align with semantically meaningful surface 3D parts. To this end, we adopt a *partition-and-parameterize* strategy based on the Shape Diameter Function (Shapira et al., 2008): we first partition the input mesh into semantic 3D parts, learn a UV parameterization for each segmented part with the same MLP-based architecture used in stage (i), and finally aggregate the per-part UV charts into a single unified UV atlas. By visibility awareness, we mean that cutting seams should be placed on 3D mesh surface regions that are unlikely to be seen under typical viewpoints and lighting, making seam artifacts less perceptible after texturing. To this end, we propose a visibility-aware objective that uses ambient occlusion as a proxy for viewer exposure (Zhukov et al., 1998), encouraging seam placement in regions of low exposure. An overview of the training procedure for *semantic awareness* and *visibility awareness* pipelines is shown in Fig. 2 and Algorithm 1, respectively.

### 3.1 PROBLEM STATEMENT AND PRELIMINARIES

Given a 3D mesh $M = (V, F)$ with vertex positions $V \in \mathbb{R}^{N_v \times 3}$ (where $N_v$ is the number of vertices), triangular faces $F \in \mathbb{R}^{N_f \times 3}$ (where $N_f$ is the number of faces), and per-vertex normals $N \in \mathbb{R}^{N_v \times 3}$, our goal is to learn a pointwise UV parameterization $u_\theta : V \to \mathbb{R}^2$ that maps each vertex in $V$ to a 2D coordinate, producing UV coordinates $Q \in \mathbb{R}^{N_v \times 2}$ with row-wise correspondence between $V$ and $Q$. We require the learned parameterization to satisfy the following properties during the learning process: *(i) geometry preservation*, i.e., the parameterization should be low-distortion and ideally preserve classical geometric properties such as (a) *isometry* (maintaining original edge lengths), (b) *equiareal* (preserving triangle/facet areas), and (c) *conformality* (preserving angles between edges); *(ii) UV-space efficiency and bijectivity*, the mapping should (a) efficiently utilize the available UV domain, (b) avoid overlapping projections of distinct 3D points, and (c) be one-to-one so that each surface point maps to a unique UV location; *(iii) high-quality cutting seams*, the parameterization should place seams in visually inconspicuous regions (Weill-Duflos et al., 2023; Ray et al., 2010; Sheffer & Hart, 2002); and *(iv) semantic awareness*, the UV mapping should semantically align 3D parts across different 3D mesh models to consistent regions in UV space to

enable cross-shape correspondence (Vermandere et al., 2024; Mukherjee et al., 2024; Cohen-Bar et al., 2025).

## 3.2 BASE NEURAL ARCHITECTURE FOR UV-PARAMETERIZATION LEARNING

To emulate the sequence of operations an artist would typically perform when constructing a UV atlas for a 3D mesh, such as stretching a canvas, placing it over the 3D surface, adding surface cuts where needed, and finally flattening it back to the plane, we build upon the idea of a deforming a 2D grid (Yang et al., 2018) by leveraging *bi-directional cycle mapping* backbone, proposed in (Zhang et al., 2024a;b; Srinivasan et al., 2024; Zhao et al., 2025). We emphasize that this stage adopts an existing previously proposed low-distortion parameterization backbone. We do not claim novelty for the backbone itself. Instead, our primary contribution lies in the design and integration semantic-aware and visibility-aware objectives as downstream tasks built on top of this established geometric foundation. Intuitively, the bi-directional cycle mapping enforces that projecting a 3D surface patch into 2D UV coordinates and re-projecting those UVs back onto the 3D mesh yields the original data. Therefore, during the learning process, any seams, distortions, or semantic mis-alignments show up as reconstruction errors. Then, by minimizing these errors, the model is guided to produce an approximately invertible and coherent parameterization that minimizes distortion. To build such a mapping, inspired by recent advances in global parameterization networks (Zhao et al., 2025; Srinivasan et al., 2024), our backbone is composed of four geometrically interpretable sub-networks: *(i) deforming network*, which starts from a regular 2D grid and adaptively deforms it into candidate UV coordinates; *(ii) wrapping network*, which maps those 2D UV candidates to the 3D surface by smoothly bending the deformed grid to conform to the input mesh, bridging the 2D and 3D domains; *(iii) cutting network*, which operates on the wrapped 3D surface to open the closed geometry by placing seams that reduce distortion for subsequent flattening; and *(iv) unwrapping network*, which projects the resulting open 3D surface back to 2D while preserving smoothness and geometric coherence. In this design, Each subnetwork is a pointwise MLP applied to mesh vertices, and the whole architecture is trained end-to-end in an unsupervised fashion. The four subnetworks form two complementary branches enforcing 2D–3D–2D and 3D–2D–3D cycles for ensuring consistency across 2D and 3D domains. Coupled with differentiable losses that encourage bijectivity and minimize distortion, this backbone produces candidate UV parameterizations for the next stages. Further details on the loss design and the interaction of the four sub-networks within the two-branch bi-directional cycle are given in Appendix A.2.

## 3.3 SEMANTIC-AWARE UV PARAMETERIZATION

The UV parameterization, obtained from the *bi-directional cycle mapping*, is used in the next step, *semantic-aware UV parameterization*. To encourage semantically coherent UV charts that simplify texture editing, transfer, and cross-object correspondence, we design a *partition-and-parameterize* pipeline (illustrated in Fig. 2). Our semantic-aware parameterization has three stages: (i) compute a per-vertex semantic partition of the input mesh using the shape diameter function (ShDF) (Shapira et al., 2008); (ii) apply the base UV-parameterization backbone from Sec. 3.2 independently to each semantic 3D part to obtain per-part UV islands; and (iii) aggregate and pack these islands into a unified UV atlas. We now describe each stage in detail.

**3D Partitioning using Shape Diameter Function.** Given an input mesh $\mathcal{M} = (V, F)$, we compute a semantic partition $S : V \to \{1, \ldots, K\}$ using the shape diameter function (ShDF) (Shapira et al., 2008) as our primary signal. Intuitively, the ShDF maps each surface sample to a scalar that estimates the local object thickness (the diameter of the object in the vicinity of that sample). Similar ShDF values typically indicate coherent semantic parts (e.g., limbs, body, handles), making the ShDF a practical cue for part-level partitioning of 3D shapes. Specifically, our 3D partitioning strategy consists of the following steps (See Appendix. A.1 for the detailed explanation of each step): *(i) compute local ShDF*, we compute a local thickness (ShDF) per face via ray casting inside a cone about the inward normal at each surface point and smoothing the samples for spatial coherence; *(ii) fit GMM*, to obtain soft per-face class likelihoods, we use a 1-D Gaussian mixture model (GMM) to fit $K$ (the deisred number of semantic components) Gaussians to the per-face thicknesses; *(iii) boundary smoothness cost*, to bias the partition toward spatially coherent regions, we then combine these likelihoods with a geometry-aware pairwise smoothness penalty (e.g., a dihedral-angle based boundary cost) on adjacent faces on the geometry to encourage coherent regions and penalize label changes across sharp edges. *(iv) initial labeling*, we then assign each face to the GMM component with maximum probability (minimum negative log-likelihood). This produces a purely data-driven segmentation that is generally noisy but captures the major mode structure of the thickness field.

*(v) refine with graph cuts*, we then refine the initial assignment by minimizing a global energy (see Eq. 10 in Appendix A.1) via iterative alpha-expansion, solving each expansion by an optimal min-cut. Repeating this process across labels and iterating a few times yields stable and low-energy 3D partitions. *(vi) Relabel connected components*, after refinement, a single label index can correspond to multiple disconnected components. To solve this issue, we relabel each connected component of faces so that every final label corresponds to a single connected sub-mesh. This guarantees that subsequent per-part processing operates on contiguous regions. *(vii) Post-processing*, to produce clean 3D segmentation results, we apply several simple post-processing heuristics to remove spurious tiny components that would complicate per-part UV parameterization. The output is then a set of connected semantic sub-meshes $\{\mathcal{M}_k = (V_k, F_k)\}_{k=1}^K$ suitable for the per-part UV parameterization stage described in Sec. 3.2. This high-level pipeline is simple, robust, and fast. Full algorithmic and implementation details are provided in Appendix A.1 for reproducibility. We also evaluate a modern zero-shot segmentation baseline, SAMesh (Tang et al., 2024), as an alternative partitioner and report comparative results in our ablation study.

---

**Algorithm 1** Visibility-Aware UV Learning

---

**Require:** Mesh $\mathcal{M} = (V, F)$ with edges $E$, vertices $V \in \mathbb{R}^{N_v \times 3}$, normals $N \in \mathbb{R}^{N_v \times 3}$, faces $F \in \mathbb{Z}^{N_f \times 3}$, per-vertex AO $\in \mathbb{R}^{N_v}$, iterations $T$, and model weights $\theta$

1: **for** $t = 1$ **to** $T$ **do**

2:    # Step 1: Learning UV map
3:    $Q \leftarrow \text{UV\_Backbone}(G, P)$ # forward pass
4:    $Q_n \leftarrow \text{normalization}(Q)$ # Norm. UVs
5:    $\mathcal{L}_{\text{wrap}} \leftarrow \text{Chamfer}(\widehat{P}, P)$ # wrap loss Eq. 16
6:    $L_{\text{unwrap}} \leftarrow \text{Rep}(Q_n)$ # Rep. loss Eq. 18

7:    Cycle-consistency loss (Eq. 17):

$$\mathcal{L}_{\text{cycle\_p}} := \|\widehat{Q} - \widehat{Q}_{\text{cycle}}\|^2 + \|P - \widetilde{P}\|^2,$$
$$L_{\text{cc\_n}} := \text{CosSim}(P_n, P_{cn})$$

8:    $e_1, e_2 \leftarrow \text{Compute\_Differential\_Properties}(P_c, Q)$
9:    $L_{\text{dist}} \leftarrow L_{\text{DDL}}(e_1, e_2) + L_{\text{TDL}}(Q, V, F)$ # Eq. 19
10:    Set base loss (Eq. 15):

$$L_{\text{base}} \leftarrow L_{\text{wrap}} + 0.01 \, L_{\text{unwrap}} + 0.01 \, L_{\text{cc\_p}} +$$
$$0.005 \, L_{\text{cc\_n}} + 0.01 \, L_{\text{conf\_diff}} + 10^{-5} \, L_{\text{conf\_tri}}$$

11:    # Step 2: Optimizing Surface Cuts
12:    Differentiable neighbor selection (Eq. 5):

$$\eta_i \leftarrow \frac{1}{\gamma} \log\left( \sum\nolimits_{j \in \mathcal{N}_i^{J_{\text{cut}}}} \exp(\gamma \|q_i - q_{i,j}\|_2) \right)$$

13:    Differentiable seam extraction (Eq. 5):
     $s_i \leftarrow \sigma(\beta(\eta_i - \tau))$
14:    Compute AO values (Eq. 4):
     $AO(p) = \frac{1}{\pi} \int_{\Omega^+(p)} V(p, \omega) \, (n(p) \cdot \omega) \, d\omega$
15:    $S \leftarrow \sum_{i \in V} s_i + \epsilon$
16:    $L_{\text{AO}} \leftarrow \frac{1}{S} \sum_{i \in V} s_i \cdot AO_i$   # seam loss (Eq. 7)
17:    $L_{\text{vis}} \leftarrow L_{\text{base}} + 0.004 \cdot L_{\text{AO}}$   # visibility loss (Eq. 8)

18:    $g \leftarrow \nabla_\theta L_{\text{vis}}(\theta)$
19:    $\theta \leftarrow \theta - \eta g$

20: **end for**
21: **return** $\theta$

---

**Per-part Parameterization.** For each semantic submesh $\mathcal{M}_k$, obtained from previous step, we instantiate the base parameterization backbone, described in Sec. 3.2 and Appendix A.2, and optimize a part-specific mapping $u_{\theta_k} : V_k \rightarrow \mathbb{R}^2$. Each part is trained with the same suite of differentiable objectives used for the global model. Therefore, we minimize the per-part loss

$$\mathcal{L}_{\text{part}}^{(k)}(\theta_k) = \mathcal{L}_{\text{wrap}}^{(k)} + \mathcal{L}_{\text{cycle}}^{(k)} + \mathcal{L}_{\text{repel}}^{(k)} + \mathcal{L}_{\text{dist}}^{(k)} \quad (1)$$

where $\mathcal{L}_{\text{wrap}}^{(k)}$, $\mathcal{L}_{\text{cycle}}^{(k)}$, $\mathcal{L}_{\text{repel}}^{(k)}$, and $\mathcal{L}_{\text{dist}}^{(k)}$ denote the wrapping objective (Eq. 16), cycle-consistency regularizer (Eq. 17), anti-overlap penalty (Eq. 18), and distortion penalty (Eq. 19), respectively (see Appendix A.2 for definitions); each term is evaluated on the submesh $\mathcal{M}_k$. To minimize Eq. 1, we compute the gradient $\nabla_{\theta_k} \mathcal{L}_{\text{part}}^{(k)}(\theta_k)$ for the bi-directional cycle mapping subnetwork of part $k$ and apply a gradient-descent update using $\theta_k \leftarrow \theta_k - \eta \nabla_{\theta_k} \mathcal{L}_{\text{part}}^{(k)}(\theta_k)$ where $\eta$ is the learning rate. This yields $K$ independently trained bi-directional sub-networks (one per semantic 3D part produced by our ShDF segmentation). The resulting per-part parameterizations are then passed to the aggregation-and-packing stage to form a unified UV atlas. Training parts independently allows each semantic region to adopt a chart that best trades off local geometric fidelity and developability.

**Atlas Aggregation and Packing.** After optimizing all parts, we obtain per-part UV islands $\{Q_k \subset [0,1]^2\}_{k=1}^K$ (each normalized to the unit square) corrsponding to each 3D segmented part.

These islands are merged into a single atlas by an *atlas aggregator*. For clarity and reproducibility our current aggregator is intentionally simple and deterministic: the final UV sheet is a unit square subdivided into a $G \times G$ regular grid, where $G = \lceil \sqrt{K} \rceil$. We then assign each part $k$ a unique grid cell indexed by row and column $(r_k, c_k)$ in row-major order. Each normalized island $Q_k$ is placed into its assigned cell via a uniform similarity transform (scale $s$ and translation $t_{r_k, c_k}$):

$$T_k(u) = s \cdot u + t_{r_k, c_k}, \qquad s = \frac{1 - 2 \cdot \text{pad}}{G}, \qquad t_{r,c} = \left[\frac{c + \text{pad}}{G} \quad \frac{r + \text{pad}}{G}\right]^T, \quad (2)$$

where $\text{pad} \in [0, 0.5)$ is a small margin (in normalized UV units) to prevent bleeding between neighboring cells. Using the same scale $s$ for all charts enforces a consistent texel density across

parts, which simplifies texture baking and editing workflows. The final per-vertex UV for the whole mesh is therefore

$$u_{\text{final}}(v) = T_k\big(u_{\theta_k}(v)\big), \qquad v \in V_k. \tag{3}$$

The grid-based aggregator is simple, deterministic, and effective for per-part editing and texture transfer, and can be replaced by more advanced packing solvers (heuristic or optimization-based) without changing the per-part losses in Eq. 1. Overall, the proposed *partition-and-parameterize* design preserves the geometry-focused objectives of Sec. 3.2 while producing semantically organized UV atlases that are easier to edit and use for correspondence.

### 3.4 VISIBILITY-AWARE UV PARAMETERIZATION

A common perceptual failure of UV atlases is that seams positioned on exposed, well-lit, or frequently viewed regions create visible discontinuities once textures are applied. To address this, we introduce a *visibility-aware* objective that biases seam placement toward surface areas with low viewer exposure. As a proxy for visibility, we leverage *ambient occlusion* (AO) (Zhukov et al., 1998), a standard geometric measure of how much of the local hemisphere around a surface point is blocked by nearby geometry (Landis, 2002; Christensen, 2002; Bavoil & Sainz, 2008; McGuire, 2010; Laine & Karras, 2010; Méndez-Feliu & Sbert, 2009). Our training procedure consists of four steps: (i) compute per-vertex AO values on the input mesh, (ii) apply the base UV-parameterization backbone (Sec. 3.2) to obtain candidate UV islands, (iii) extract UV boundary points corresponding to cutting seams, and (iv) compute and minimize the AO-weighted average over the associated 3D vertices. This objective guides the backbone to produce UV islands whose seams coincide with regions of low exposure, thereby relocating cuts to less visible areas of the 3D surface and reducing perceptible seam artifacts after texturing and rendering. We now detail each step in turn.

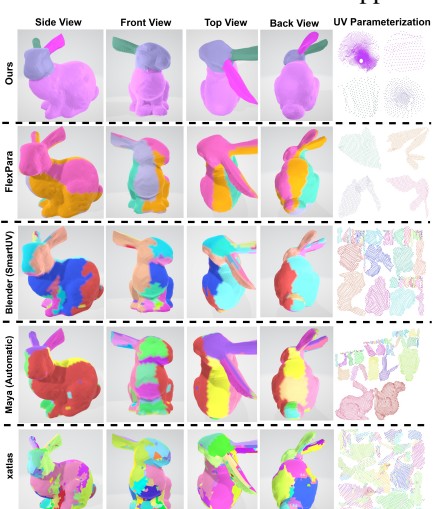

Figure 3: Qualitative results of the proposed semantic-aware UV parameterization method on a Rabbit mesh. For each method, we show the rendered 3D object from multiple viewpoints, with the corresponding UV atlas in the rightmost column. As shown, our method produces UV charts that are align more semantically with the mesh's 3D semantic parts, unlike the baselines.

**Compute Per-Vertex Ambient Occlusion.** Let $p$ be a point on the surface with unit normal $n(p)$. Define the hemisphere of outgoing directions as $\Omega^+(p) = \{\omega \in \mathbb{S}^2 : n(p) \cdot \omega > 0\}$. The binary visibility function $V(p, \omega) =$

$$\begin{cases} 1, & \text{if the ray starting at } p \text{ in direction } \omega \text{ does not intersect the surface (unoccluded)} \\ 0, & \text{otherwise (occluded)} \end{cases}$$ measures

whether direction $\omega$ is visible from $p$. The ambient-occlusion exposure $AO(p) \in [0, 1]$ is then defined as the cosine-weighted average visibility over the hemisphere $\Omega^+(p)$:

$$AO(p) = \frac{1}{\pi} \int_{\Omega^+(p)} V(p, \omega) \, (n(p) \cdot \omega) \, d\omega. \tag{4}$$

By this convention, $AO(p) = 1$ indicates a fully exposed point (no occlusion) and $AO(p) = 0$ indicates a fully occluded point. In practice AO is estimated by Monte Carlo hemisphere sampling (Veach, 1998) or by high-quality offline ray sampling (Pharr et al., 2023) for evaluation. In our implementation, we compute ambient occlusion using the libigl geometry processing library (Jacobson et al., 2018).

**Learning Geometry-Preserving Mesh Parameterization.** Once per-vertex AO values are computed, we generate the UV parameterization using the base backbone described in Sec. 3.2 and Appendix. A.2 to obtain a potential UV parameterization candidate.

**Cutting Seam Extraction in UV Space.** We then extract cutting seams directly from the UV mapping learned in the previous step. This process has three steps: *(i) neighbor selection*, given a set of 3D surface points $P$ with their corresponding UV coordinates $Q$, the method determines whether each vertex lies on a seam by exploring its neighborhood. For each 3D point $p_i \in P$

with UV coordinate $q_i \in Q$, we first find its $N$ nearest neighbors in 3D, denoted $\{p_{i,j}\}_{j=1}^N$, with corresponding UV coordinates $\{q_{i,j}\}_{j=1}^N$. *(ii) differentiable max UV distance computation,* we then (similar to Zhao et al. (2025)) compute the maximum UV distance between $q_i$ and its neighbors:

$$\eta_i = \max\left(\|q_i - q_{i,j}\|_2\right)_{j=1}^N \tag{5}$$

Next, we define a threshold $\tau$ based on the UV domain size. Specifically, we compute the bounding square of the UV map with side length $L(Q)$, and set $\tau = \tau_{\text{scale}} \cdot L(Q)$ where $\tau_{\text{scale}}$ is a constant (e.g., 0.1 by default). The intuition behind Eq. 5 is that, during training, at each iteration we must identify the cutting-seam (boundary) points in UV space in order to evaluate their ambient occlusion values and then compute the visibility-aware loss in Eq. 7. Minimizing this loss encourages the model to learn UV parameterizations whose seams lie in less-visible (low-AO) regions. Eq. 5 provides a mechanism for detecting these seam points. Rather than locating seams directly on the 3D mesh, we detect them in the UV domain based on how local neighborhoods deform during unwrapping. Specifically, for each 3D vertex we know both its 3D coordinates and its corresponding UV coordinates at the current iteration. Therefore, we examine the 3D neighbors of each vertex and compare their mapped UV positions. If a pair of vertices are close neighbors on the 3D surface but become separated by more than a threshold distance in UV space, this indicates that the surface has been cut between them. We therefore mark such UV points as seam (boundary) points. This criterion is grounded in the fact that any valid unwrapping must introduce discontinuities (i.e., seams): vertices that are adjacent on the 3D surface become far apart in the UV plane after the mesh is cut open. Eq. 5 formalizes this idea by detecting precisely those locations where this large separation occurs. However, this maximization problem in Eq. 5 is not inherently differentiable, so we introduce mathematical modifications to ensure differentiability during the learning process. Specifically, instead of a hard decision, we assign each vertex a differentiable *soft seam membership* score: $s_i = \sigma\big(\beta\,(\eta_i - \tau)\big)$ where $\sigma(\cdot)$ is the sigmoid function and $\beta$ controls the sharpness of the transition. The maximization problem can then be formulated as a differentiable function:

$$\eta_i \approx \frac{1}{\gamma}\log\Big(\sum_{j\in\mathcal{N}_i^N}\exp\big(\gamma\,\|q_i - q_{i,j}\|_2\big)\Big), \qquad s_i = \sigma\big(\beta(\eta_i - \tau)\big), \qquad \tau = \tau_{\text{scale}}\,L(Q) \tag{6}$$

where $\mathcal{N}_i^N$ denotes the $N$ nearest 3D neighbors in the 1-ring of vertex $i$, and $\gamma, \beta, \tau_{\text{scale}}$ control the soft-max / sigmoid sharpness and threshold scale. With this formulation, vertices with $s_i$ close to 1 are highly likely to lie on seams, while vertices with $s_i$ near 0 are not. This procedure produces a soft mask $s \in [0,1]^V$ that marks the cutting seams in UV space and is used in the next step to compute the ambient occlusion loss on the seam vertices.

**Visibility-Aware Seam Loss.** After identifying soft seam memberships $s_i \in [0,1]$ for each vertex $v_i$, we encourage seams to lie in less visible regions of the surface. As a visibility proxy, we use per-vertex ambient occlusion values $AO_i \in [0,1]$, computed in the first step, where 0 indicates fully occluded and 1 indicates

Table 1: Quantitative comparison of the proposed visibility-aware UV parameterization method against baselines on multiple evaluation metrics.

| Method | Visibility ↓ | Conformality ↑ | Equiareality ↑ | Time (s) ↓ |
|---|---|---|---|---|
| OptCuts (Li et al., 2018) | 0.7855 | **0.9341** | **0.8934** | 240 |
| FlexPara (Single-Chart) (Zhao et al., 2025) | 0.8604 | 0.9097 | 0.6759 | 2 |
| Ours - Visibility-Aware Param | **0.6065** | 0.9175 | 0.6093 | 2 |
| Ours - Semantic+Visibility-Aware Param | 0.6534 | 0.9153 | 0.6369 | 15 |

fully visible. The loss is defined as the weighted average of occlusion values, using soft seam memberships as weights:

$$\mathcal{L}_{\text{AO}} = \frac{\sum_i s_i\,AO_i}{\sum_i s_i + \varepsilon} \tag{7}$$

where $\varepsilon$ is a small constant to avoid division by zero. Intuitively, if the network assigns high seam weight $s_i$ to highly visible vertices (high $AO_i$), the loss is large. Conversely, assigning seam weight to occluded vertices (low $AO_i$) reduces the loss. Therefore, minimizing $\mathcal{L}_{\text{AO}}$ guides seam placement toward regions of low exposure.

**Integration with The Backbone Losses.** To jointly optimize geometric and visibility properties, we augment the visibility-aware seam loss $\mathcal{L}_{\text{AO}}$ with the backbone objective from Sec. 3.2 and Appendix A.2. The full visibility-aware objective is then becomes

$$\mathcal{L}_{\text{vis}}(\theta) = \mathcal{L}_{\text{wrap}} + \mathcal{L}_{\text{cycle}} + \mathcal{L}_{\text{repel}} + \mathcal{L}_{\text{dist}} + \lambda_{\text{vis}}\,\mathcal{L}_{\text{AO}} \tag{8}$$

where $\mathcal{L}_{\text{wrap}}, \mathcal{L}_{\text{cycle}}, \mathcal{L}_{\text{repel}}$, and $\mathcal{L}_{\text{dist}}$ denote the wrapping objective (Eq. 16), cycle-consistency regularizer (Eq. 17), anti-overlap penalty (Eq. 18), and distortion penalty (Eq. 19), respectively (see Appendix A.2 for definitions). To minimize Eq. 8, we compute the gradient $\nabla_\theta \mathcal{L}_{\text{vis}}(\theta)$ for

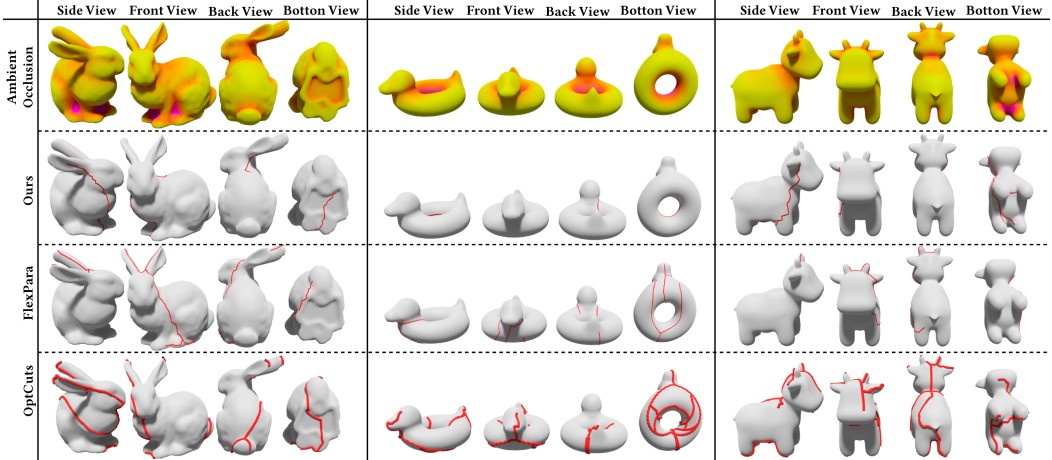

Figure 4: Qualitative results for visibility-aware seam placement and UV parameterization on three representative meshes. For each mesh, the top row shows per-vertex ambient occlusion (yellow = exposed, purple = occluded). Beneath are the visualizations of cutting seams (red) from our method, FlexPara, and OptCuts (top to bottom). Our method places a larger fraction of seam geometry in less-exposed regions, reducing the likelihood of visible seam artifacts under typical viewpoints.

the bi-directional cycle mapping network, and then apply a gradient-descent update using $\theta \leftarrow \theta - \eta \nabla_\theta \mathcal{L}_{\text{vis}}(\theta)$ where $\eta$ is the learning rate.

## 4 EXPERIMENTS AND RESULTS

We evaluate our unsupervised semantic- and visibility-aware UV parameterization pipelines on a diverse collection of meshes and compare against several baselines. We provide both qualitative and quantitative comparisons to: FlexPara (Zhao et al., 2025) (recent state-of-the-art global parameterization network), Autodesk Maya (Autodesk, Inc., 2025) (industry-grade commercial software), Blender (Blender Foundation, 2025) (popular open-source modeling tool), and xatlas (jpcy, 2025) (widely-used C++ library for generating unique texture coordinates suitable for lightmap baking and texture painting). Figs. 1, 3, 4, and 5 show representative qualitative results, and Tables. 1 and 3 presents quantitative results of our semantic-aware and visibility-aware pipelines against baselines (additional results are provided in Appendix A.3). In the following, we detail each experiment and its corresponding results.

### 4.1 QUALITATIVE EVALUATION

**Comparative Evaluation of Semantic-Aware UV Parameterization.** To evaluate the performance of the proposed semantic-aware UV parameterization pipeline, we compare our method against FlexPara, Autodesk Maya's Automatic UV generator, Blender's SmartUV, and xatlas. Fig. 3 shows a representative

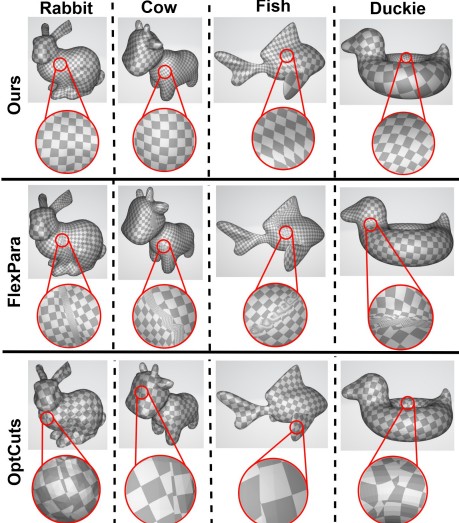

Figure 5: Checkerboard texturing using UV maps produced by our visibility-aware method, FlexPara, and OptCuts. Each row shows rendered views of different meshes textured with a checkerboard and a magnified inset of a visually important region near seams (red circles). Because our method steers seams toward occluded regions, the checkerboard pattern appears substantially more continuous from typical camera viewpoints. By contrast, baselines exhibit visible seam artifacts in the zoomed-in insets.

comparison on a single Bunny mesh; additional results on more diverse and complex meshes are provided in Figs. 11 and 12 in Appendix A.3. In Fig. 3 and Fig. 11, each row corresponds to one method. Semantic parts are color-coded, and columns show the rendered 3D mesh from multiple camera viewpoints. The rightmost column depicts the unwrapped UV coordinates, where UV islands are color-coded consistently with their corresponding semantic 3D parts in the rendered views. As shown, our method successfully identifies semantic parts of the input mesh, parameterizes them, and maps them into UV space with low distortion. This leads to cleaner, more interpretable charts

that directly benefit downstream tasks such as texture painting and transfer. By contrast, the baseline methods fail to consistently capture and parameterize semantic parts of the 3D mesh.

**Comparative Evaluation of Visibility-Aware UV Parameterization.** To evaluate the performance of the proposed visibility-aware UV parameterization pipeline, we compare our method against FlexPara (Zhao et al., 2025) and OptCuts (Li et al., 2018). Figs. 4 and 8 present representative comparisons on different meshes. Additional examples appear in Figs. 5, 9, and 10 in Appendix A.3. In Fig. 4 the top row visualizes per-vertex ambient occlusion (AO): lighter tones (e.g., yellow) indicate high exposure (more visible) while darker tones (e.g., purple) indicate low exposure (more occluded). A desirable parameterization therefore places cutting seams on darker, less-visible regions to reduce perceptible seam artifacts after texturing and rendering. The rows beneath show UV parameterizations produced by our method, FlexPara, and OptCuts (in that order). Qualitatively, our method allocates a larger fraction of seam geometry to low-exposure regions compared to the baselines. Table 1 (and Fig. 7 in Appendix A.3) quantitatively supports this observation: the report the mean AO values of seam vertices for each method, where lower scores indicate seams located in less-visible (more occluded) areas. As shown, our method consistently attains lower mean seam-AO, confirming that the visibility-aware objective successfully biases seam placement toward less-exposed surface regions. Also, Fig. 5 illustrates checkerboard texturing on different mesh objects using parameterizations from our method and from baselines. As shown, our method produces noticeably more seamless texture patterns when rendered from visible viewpoints, whereas FlexPara and OptCuts exhibit prominent cutting seams (highlighted in the zoomed-in insets of Fig. 5).

## 4.2 QUANTITATIVE EVALUATION

**User Study.** To evaluate semantic- and visibility-awareness of the our method, we conducted a user-preference study with 115 participants. We distributed the questionnaire to two groups: experts and general participants. Of the 115 participants, 70 are general participants (including graduate students with computer science and engineering backgrounds) and 45 are experts: 31 software engineers, 3 project managers, 2 product owners, 5 UV/layout artists, and 4 modeling artists, all working in the Media and Entertainment industry for film and games. Each participant completed 11 comparisons between textured 3D shapes and UV maps produced by our method, FlexPara, OptCuts, Autodesk Maya, Blender, and xatlas. For each comparison, participants were rated each result according to three visual criteria: (i) texture-pattern smoothness/continuity, (ii) semantic alignment (how well colors correspond to meaningful parts), and (iii) seam visibility (how well seams are placed in occluded/less-obvious regions). The semantic-aware and visibility-aware evaluations comprised 5 and 6 questions, respectively, for a total of 11 comparisons per participant. Table 2 reports the percentage of expert participant preferences for each method. As the table shows, expert users, particularly UV, layout, and modeling artists, strongly prefer our proposed method over the baselines. Examples of questions in our user-study questionnaire are shown in Fig. 15. Also, Table 10 in Appendix A.3 reports the percentage of general participant preferences for each method. As shown in Table 10, general users also strongly prefer our proposed method over the baselines.

Table 2: To evaluate semantic- and visibility-awareness of the proposed method, we conducted a user-study with 45 expert participants performing 11 comparisons between textured 3D shapes and UV parameterizations produced by our method and baselines. Our proposed method is strongly preferred by the expert users over the baselines.

| Method | Expert User Preference Percentage |
|---|---|
| **Visibility-Awareness Evaluation** | |
| OptCuts (Li et al., 2018) | 1.48 % |
| FlexPara (single-Chart) (Zhao et al., 2025) | 4.81 % |
| Ours - Visibility-Aware Param | **93.70 %** |
| **Semantic-Awareness Evaluation** | |
| xatlas (jpcy, 2025) | 2.66 % |
| Blender (Blender Foundation, 2025) | 5.33 % |
| Autodesk Maya (Autodesk, Inc., 2025) | 8.44 % |
| FlexPara (multi-Chart) (Zhao et al., 2025) | 3.33 % |
| Ours - Semantic-Aware Param | **80.22 %** |

**Semantic- and Visibility-Awareness Evaluation.** We also quantitatively evaluate both the semantic-aware and visibility-aware components of our pipeline on a collection of representative meshes. For the semantic-aware pipeline we report (i) *Hamming distance* (lower is better), the fraction of vertices whose labels disagree after optimal label matching to the reference (we use SAMesh (Tang et al., 2024) outputs as a ground-truth); and (ii) *Rand Index (RI)* (higher is better), a pairwise agreement measure that counts how often vertex pairs are grouped or separated identically between the two labelings. For the *visibility-aware* pipeline we report *Mean seam AO* (lower is better in our AO convention, where 0 = occluded and 1 = exposed), the average ambient-occlusion value over seam vertices, which indicates how well seams are placed in occluded regions. Tables 1 and 3 summarize these metrics and compare our method against the baselines described in Sec. 4. Further discussion of the results in these two tables are provided in Appendix A.3.

**Geometry-Preservation Evaluation.** In addition to task-specific metrics mentioned above, in Tables 1 and 3, we evaluate geometric quality for all methods using *conformality (angle-preservation)* and *equiareality (area-preservation)* (higher is better) to ensure the parameterizations remain faithful enough to the original surface. The proposed semantic-aware and visibility-aware UV parameterization frameworks produce an increase in standard distortion metrics including angle-distortion and area-distortion. This is a trade-off which is expected when optimizing multiple objectives during the learning process. Specifically, our optimization introduces explicit semantic-alignment and visibility-awareness terms that encourage semantically consistent charts and reduce conspicuous seams. However, existing geometry-preserving methods (e.g., Autodesk Maya, Blender, xatlas, and FlexPara) focus only on traditional distortion objectives and therefore score better on those numbers, but they do not provide the semantic- and visibility-related guarantees produced by our method. We note that these semantic and visibility properties are often more valuable in some downstream tasks such as mass production, semantic editing, and 3D asset reuse, where consistent UV semantics and low seam visibility

Table 3: Quantitative comparison of the proposed semantic-aware UV parameterization method against baselines on multiple evaluation metrics.

| Method | Semantic Awareness | | Conformality ↑ | Equiareality ↑ | Inference Time (sec) ↓ |
|---|---|---|---|---|---|
| | Hamming Distance ↓ | Rand Index ↑ | | | |
| xatlas (jpcy, 2025) | 0.8896 | 0.7023 | **0.9792** | **0.9341** | 12 |
| Blender (Blender Foundation, 2025) | 0.8634 | 0.7173 | 0.9289 | 0.9023 | < 1 |
| Maya (Autodesk, Inc., 2025) | 0.8615 | 0.7125 | 0.9272 | 0.8746 | < 1 |
| FlexPara (Multi-Chart) (Zhao et al., 2025) | 0.5980 | 0.6902 | 0.9592 | 0.7606 | 2 |
| Ours - Semantic-Aware Param | **0.3188** | **0.8151** | 0.9123 | 0.6707 | 15 |
| Ours - Semantic+Visibility-Aware Param | 0.3212 | 0.8087 | 0.9153 | 0.6369 | 15 |

significantly reduce manual effort. Importantly, the trade-off is controllable in our framework: the relative emphasis between distortion and semantic/visibility objectives can be controlled by tunable weights and hyperparameters between different loss functions in Equations (1) and (8), so users can shift toward lower distortion when that is preferred.

**Ablation Studies.** To understand which design choices drive our results, we perform a set of controlled ablations and summarize the key findings below (additional qualitative examples are in Fig. 13 and Fig. 14 in Appendix A.3). *(i) Partitioning strategy (ShDF vs. SAMesh).* We replace our default Shape Diameter Function (ShDF) partitioner (Shapira et al., 2008) with SAMesh (Tang et al., 2024) while keeping the remainder of the *partition-and-parameterize* pipeline identical. Qualitatively both partitioners produce part decompositions that align with geometric structure across simple and complex models, resulting compact, low-distortion per-part UV charts (see the appendix Fig. 13 and Fig. 14). Quantitatively (Table 6 in Appendix. A.3), using ShDF in our pipeline yields UV parameterizations that are more conformal (better angle preservation) and more equiareal (better area preservation) than those produced when using SAMesh. *(ii) Visibility loss (with vs. without).* We ablate the proposed visibility-aware objective (Sec. 3.4) by training identical models with and without $\mathcal{L}_{\text{vis}}$. The visibility-aware objective consistently lowers the mean AO of seam vertices (Fig 7 in Appendix. A.3), i.e., it moves seams to regions of lower exposure (lower AO values). Qualitatively, models trained with the AO loss produce seams that are less salient in rendered images from canonical viewpoints (see Fig. 5). Importantly, as presented in Table. 1, adding the visibility term does not noticeably degrade geometric distortion metrics, conformality (angle-preservation) and equiareality area-preservation). The ablations show that: (i) the ShDF partitioner is a semantically reliable and geometry-driven initializer for our semantic-aware parameterization across diverse shapes; and (ii) the visibility-aware loss is required to effectively move seams to less-exposed regions without harming geometric quality. Additional ablation studies are provided in Appendix A.3.

## 5 Conclusions

This study presents an unsupervised representation-learning framework for semantic- and visibility-aware UV parameterization of 3D meshes. The semantic-aware pipeline leverages high-level mesh partitioning to align UV islands with meaningful 3D parts, producing UV charts that are easier to interpret and manipulate for downstream tasks such as texture synthesis. The visibility-aware pipeline uses ambient occlusion as a differentiable proxy for visual exposure to guide seam placement toward less-visible regions, reducing perceptual artifacts in texturing and rendering. Extensive qualitative and quantitative comparisons to state-of-the-art methods show our parameterizations achieve low distortion, stronger semantic alignment, and seams placed in perceptually favorable locations. We expect these contributions to serve as building blocks for generative 3D models and broader 3D content creation. Therefore, future work includes jointly learning UV parameterization and texture generation.

ACKNOWLEDGMENTS

This research was supported by Autodesk Research and the Autodesk AI Lab. We are especially thankful for the collaborative environment, technical resources, and insightful discussions that contributed to shaping this work.

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

# A APPENDIX

## A.1 REMARKS ON SHAPE DIAMETER FUNCTION (SHDF)

Our semantic-aware parameterization, explained in Sec.3.3, relies on computing a per-vertex semantic partition of the input mesh using shape diameter function (Shapira et al., 2008). To estimate per-vertex semantic label, we design and implement a method based on Shape Diameter Function (Shapira et al., 2008) that involves the following key steps.

**Compute a per face local thickness (shape-diameter) field.** For a surface sample $p$ (face centroid or vertex), the value $\text{ShDF}(p)$ estimates the local object diameter by probing the interior along a cone centered on the inward normal at $p$. Let $\mathcal{R}(p) = \{r_i\}_{i=1}^R$ be a set of rays radiating from $p$ inside a cone of half-angle $\alpha$ around the inward normal. For each ray $r_i$ we compute the length $\ell_i$ of the intersection segment between $r_i$ and the mesh interior (i.e., the distance from $p$ to the first opposite intersection). We robustly aggregate the ensemble $\{\ell_i\}$ using a median (or trimmed mean) to obtain: $\text{ShDF}(p) = \text{median}\left(\{\ell_i\}_{i=1}^R\right)$. This median-based aggregation reduces sensitivity to spurious long or short intersections caused by complex local geometry. In practice, to estimate a scalar per surface element that measures local object thickness (intuitively, it presents the diameter of the object in the neighborhood of each sample), we use ray casting (using PyMeshLab (Muntoni & Cignoni, 2021)) in multiple directions per sample and return a per-face scalar. Then, the raw scalars are normalized to $[0, 1]$ and re-scaled via a monotonic compression (e.g., a log-like transform) to reduce dynamic range and improve cluster separability. In our implementation we use per-face ShDF evaluated at face centroids with the following defaults: cone angle $\theta = 120°$ (as in the original work (Shapira et al., 2008)), number of rays $R = 60$, and ray sampling distributed uniformly within the cone.

**Smoothing and normalization.** Similar to (Shapira et al., 2008), we smooth raw ShDF samples on the mesh to promote spatial coherence and reduce pose-dependent noise. We apply a small number (e.g., 1-2) of iterations of Laplacian smoothing on the ShDF scalar field with area-weighted averaging. To balance values across scales and to boost contrast of small parts, we perform a normalization:

$$\tilde{\text{ShDF}}(p) = \frac{\log(1 + \text{ShDF}(p)) - \min_p \log(1 + \text{ShDF}(p))}{\max_p \log(1 + \text{ShDF}(p)) - \min_p \log(1 + \text{ShDF}(p))}. \tag{9}$$

**Fit a Gaussian Mixture Model (GMM) to ShDF values.** Given a desired number of semantic components, to obtain soft class probabilities, we fit a one-dimensional Gaussian mixture model (GMM) to the per-face thickness scalars obtained from previous step. The probabilities from the GMM provide a data-driven per-face likelihood for each component and serve as the unary data term in the subsequent energy formulation.

**Construct a boundary smoothness cost between adjacent faces.** To bias the partition toward spatially coherent regions, we compute a smoothness cost on adjacent face pairs based on local geometry (for example, a function of the dihedral angle). Intuitively, nearly-coplanar adjacent faces incur a low boundary penalty, whereas faces separated by sharp edges incur a higher penalty for being assigned the same label.

**Initial partition assignment via data likelihood.** As an initial labeling, each face is assigned to the GMM component with maximum probability (equivalently, minimum negative log-likelihood of the probability). This produces a purely data-driven segmentation that is generally noisy but captures the major mode structure of the thickness field.

**Refine partitioning via energy-based graph cuts.** The initial assignment, obtained from the previous step, is then refined using an iterative expansion-based graph-cut procedure that minimizes a global energy

$$E(\text{labeling}) = \sum_{\text{faces}} \mathcal{C}_{\text{data}}(\text{face}, \text{label}) + \sum_{\text{adjacent faces}} \mathcal{C}_{\text{smooth}}(\text{pair}) \cdot \mathbf{1}[\text{labels differ}], \tag{10}$$

where the unary term is derived from the GMM negative log-likelihoods and the pairwise term is the smoothness cost described above. The expansion moves are solved by an optimal min-cut on an auxiliary graph for each label. Repeating this process across labels and iterating a few times yields a stable and low-energy partition.

**Re-label connected components.** After refinement, a single label index can correspond to multiple disconnected components. To solve this issue, we relabel each connected component of faces so that

every final label corresponds to a single connected sub-mesh. This guarantees that subsequent per-part processing operates on contiguous regions.

**Post-processing.** To produce clean 3D segmentation results, we apply simple post-processing heuristics: (i) remove tiny components below a face-count threshold, (ii) smooth labels by majority filtering on local neighborhoods, and (iii) (when appropriate) merge adjacent labels that are semantically similar. These operations stabilize the partition and remove spurious tiny regions that would complicate per-part UV parameterization. The output is then a set of connected semantic sub-meshes $\{\mathcal{M}_k = (V_k, F_k)\}_{k=1}^{K}$ suitable for the per-part UV parameterization stage described in Sec.3.2.

## A.2 REMARKS ON BASE ARCHITECTURE FOR UV-PARAMETERIZATION LEARNING

**Overview.** Our backbone, based on bi-directional cycle mapping proposed in (Zhao et al., 2025), implements a learnable and interpretable pipeline that mimics the artist operations used to create a UV atlas: *(i) deforming network*, starting from a regular 2D grid of uniformly spaced coordinates, this network adaptively deforms the grid to produce a set of candidate UV coordinates that are potentially well-suited for mapping; *(ii) wrapping network*, this module maps 2D UV candidates, generated by the deforming network, to the 3D surface by smoothly bending and folding the deformed planar grid until it conforms to the target geometry (in this case the input 3D mesh) which leads to effectively bridge the 2D and 3D domains; *(iii) cutting network*, operating directly on the 3D surface, generated by the wrapping network, this network transforms the input closed 3D geometry, obtained from the wrapping network, into an open, more developable 3D surface manifold. By strategically determining seam locations, i.e. where to cut the input closed 3D surface, it facilitates distortion reduction in subsequent flattening steps; and *(iv) unwrapping network*, this final network projects the open 3D surface points back to the 2D domain, preserving smoothness and geometric coherence. In this design, each subnetwork is designed as a multilayer perceptron (MLP) applied pointwise to mesh vertices. The entire architecture is trained jointly in an end-to-end unsupervised fashion. The four subnetworks are assembled into two complementary branches that enforce both 2D-3D-2D and 3D-2D-3D cycle mappings, ensuring consistency across 2D and 3D domains. Coupled with differentiable loss functions designed to encourage bijectivity and minimize distortion, this backbone forms the base architecture of our method to achieve a candidate UV parameterization for the next stages. In the following, we detail on the loss designs and the interaction of the four sub-networks within the two-branch bi-directional cycle.

**Notation.** Let $\mathcal{M} = (V, F)$ be a triangle mesh with vertex positions $P \in \mathbb{R}^{V \times 3}$ and normals $N \in \mathbb{R}^{V \times 3}$. Let $G = \{g_i\}_{i=1}^{V} \subset \mathbb{R}^2$ denote a canonical planar lattice (the artist's canvas). The learnable UV mapping is $u_\theta(\cdot)$; the four MLP sub-networks are $DeformNet, WrapNet, CutNet, UnwrapNet$.

**Sub-network Definitions.** Each module is implemented as a small pointwise MLP. We use additive residuals in the deform and cut modules to bias predictions toward small, interpretable offsets:

$$\widehat{Q} = DeformNet(G) = G + \phi_d''([\phi_d'(G); G]), \tag{11}$$

$$[\widehat{P}, \widehat{N}] = WrapNet(\widehat{Q}) = \phi_w''([\phi_w'(\widehat{Q}); \widehat{Q}]), \tag{12}$$

$$\widehat{P}_{\text{cut}} = CutNet(\widehat{P}) = \widehat{P} + \phi_c''([\phi_c'(\widehat{P}); \widehat{P}]), \tag{13}$$

$$\widehat{Q}_{\text{cycle}} = UnwrapNet(\widehat{P}_{\text{cut}}) = \phi_u(\widehat{P}_{\text{cut}}). \tag{14}$$

where $\widehat{Q}$ are deformed 2D candidates, $\widehat{P}$ are wrapped 3D points (with normals $\widehat{N}$), $\widehat{P}_{\text{cut}}$ is the opened 3D surface after the learned cut offsets, $\widehat{Q}_{\text{cycle}}$ are the re-flattened coordinates produced by the UnwrapNet, $\phi_d'' : \mathbb{R}^{(h+2)} \to \mathbb{R}^2$, $\phi_d' : \mathbb{R}^2 \to \mathbb{R}^h$, $\phi_w' : \mathbb{R}^2 \to \mathbb{R}^h$, $\phi_w'' : \mathbb{R}^{h+2} \to \mathbb{R}^6$, $\phi_c'' : \mathbb{R}^{h+3} \to \mathbb{R}^3$, $\phi_c' : \mathbb{R}^3 \to \mathbb{R}^h$, and $\phi_u : \mathbb{R}^3 \to \mathbb{R}^2$ are implemented as stacked MLP layers.

**Sub-network architectures.** Each subnetwork is implemented as a sequence of PWE blocks (pointwise 1×1 convolution with LeakyReLU activation functions), applied point-wisely to per-vertex features. In practice each MLP is split into an encoding stage (MLP_1) and a decoding stage (MLP_2) (e.g., Deform use $[2 \to 512 \to 512 \to 512 \to 64]$ then $[66 \to 512 \to 512 \to 512 \to 2]$ as encoder and decoder, respectively). DeformNet produces deformed UV candidates, WrapNet maps those

Table 4: Layer-wise summary of four sub-networks used in our base neural UV architecture (PWE = point-wise embedding conv1d). Each listed MLP is a sequence of PWE blocks; intermediate PWEs use LeakyReLU (with the negative slope of 0.01) while the final PWE is linear.

| Subnetwork | Component | Layer channels | # PWE | Input / Output dim | Notes (activation) |
|---|---|---|---|---|---|
| DeformNet | MLP_1 | $[2 \to 512 \to 512 \to 512 \to 64]$ | 4 | 2 / 64 | LeakyReLU(0.01) on hidden PWEs; final PWE linear |
| | MLP_2 | $[66 \to 512 \to 512 \to 512 \to 2]$ | 4 | 66 / 2 | LeakyReLU(0.01) on hidden PWEs; final PWE linear |
| WrapNet | MLP_1 | $[2 \to 512 \to 512 \to 512 \to 64]$ | 4 | 2 / 64 | same as Deform MLP_1 |
| | MLP_2 | $[66 \to 512 \to 512 \to 512 \to 6]$ | 4 | 66 / 6 | outputs 6D (3 coords + 3 normals) |
| CutNet | MLP_1 | $[3 \to 512 \to 512 \to 64]$ | 3 | 3 / 64 | LeakyReLU(0.01) on hidden PWEs; final PWE linear |
| | MLP_2 | $[67 \to 512 \to 512 \to 3]$ | 3 | 67 / 3 | produces 3D offsets $X_o$ (then $X_c = X + X_o$) |
| UnwrapNet | MLP | $[3 \to 512 \to 512 \to 2]$ | 3 | 3 / 2 | maps cut 3D points $\to$ 2D unwrapped points |
| | CutNet | see above | — | 3 / 3 | Unwrapping uses Cutting before this MLP |
| PWE (block) | conv layer | Conv1d(kernel=1): $C_{\text{in}} \to C_{\text{out}}$ | 1 (per call) | - | point-wise 1D conv + bias; optional LeakyReLU(0.01) |

2D candidates to 3D points and normals, CutNet predicts pointwise 3D offsets to open the surface (making it more developable), and UnwrapNet applies CutNet followed by a final MLP to project the opened surface back to 2D. All subnetworks operate pointwise (per-vertex), the final PWE in each MLP is linear, intermediate PWEs use LeakyReLU as activation functions, and the entire architecture is trained end-to-end in an unsupervised manner (see Table 4 for a layer-wise breakdown).

**Bi-Directional Cycle Mapping.** We follow (Zhang et al., 2024b; Srinivasan et al., 2024; Zhao et al., 2025) to build our base neural UV parameterization architecture with two-branch MLP architecture that leverages the four sub-networks, explained above.

- **2D-3D-2D:** $G \xrightarrow{DeformNet} \widehat{Q} \xrightarrow{WrapNet} \widehat{P} \xrightarrow{CutNet} \widehat{P}_{\text{cut}} \xrightarrow{UnwrapNet} \widehat{Q}_{\text{cycle}}$. Enforcing $\widehat{Q}$ to be close to $\widehat{Q}_{\text{cycle}}$ stabilizes deformations and ensures that the learned 2D deformations are actually realizable on the surface.

- **3D-2D-3D:** $P \xrightarrow{CutNet} P_{\text{cut}} \xrightarrow{UnwrapNet} Q \xrightarrow{WrapNet} \widetilde{P}$. Enforcing $\widetilde{P}$ to be close to $P$ constrains the unwrapping map so that the predicted UV coordinates can be re-embedded to match the true geometry.

Together the two cycles expose inconsistencies inclusing any seams, overlaps, stretches, and/or distortions as reconstruction losses that the network can reduce. We now detail all the reconstruction losses in detail.

**Training losses.** The backbone is trained with a weighted sum of unsupervised, geometry-aware losses:

$$\mathcal{L} = \lambda_{\text{wrap}}\mathcal{L}_{\text{wrap}} + \lambda_{\text{cycle}}\mathcal{L}_{\text{cycle}} + \lambda_{\text{repel}}\mathcal{L}_{\text{repel}} + \lambda_{\text{dist}}\mathcal{L}_{\text{dist}} \tag{15}$$

Below are the main terms.

**Wrapping loss.** Encourages $\widehat{P}$ (Wrap-Net output) to approximate the input surface by computing the chamfer distance between the input mesh vertices $P$ and reconstructed $\widehat{P}$:

$$\mathcal{L}_{\text{wrap}} = \text{ChamferDistance}(\widehat{P}, P) + \kappa_{\text{norm}}\left(1 - \text{CosineSim}(\widehat{N}, N)\right) \tag{16}$$

where the second term enforces vertex normal consistency by computing point-wise cosine similarity between normal vectors.

**Cycle-consistency loss.** Penalizes reconstruction error on both cycles by computing the squared norm 2 of the pair of deformed 2D candidates $\widehat{Q}$ and the output of UnwrapNet $\widehat{Q}_{cycle}$ and the pair of input mesh vertices $P$ and reconstructed vertices $\widehat{P}$ from the WrapNet:

$$\mathcal{L}_{\text{cycle}} = \|\widehat{Q} - \widehat{Q}_{\text{cycle}}\|_2^2 + \|P - \widetilde{P}\|_2^2 \tag{17}$$

optionally augmented with normal/feature consistency terms.

**Anti-overlap (repulsion).** Discourages UV overlaps by repelling nearby vertex pairs in UV space:

$$\mathcal{L}_{\text{repel}} \;=\; \frac{1}{|\mathcal{N}|} \sum_{(i,j)\in\mathcal{N}} \phi\big(\|u_\theta(i) - u_\theta(j)\|_{\text{uv}}\big) \tag{18}$$

where $\mathcal{N}$ denotes local neighbor pairs, $\phi$ is a hinge-style repulsion, and $\|\cdot\|_{\text{uv}}$ includes chart-periodicity handling.

**Differential and triangle distortion losses.** Control conformal and area distortion using a Jacobian-level differential loss (DDL) and a triangle-based discrete loss (TDL):

$$\mathcal{L}_{\text{dist}} = \lambda_{\text{DDL}}\, \mathcal{L}_{\text{DDL}} + \lambda_{\text{TDL}}\, \mathcal{L}_{\text{TDL}} \tag{19}$$

The DDL is computed by applying automatic differentiation to the MLP outputs to approximate the local Jacobian, while the TDL explicitly measures changes in angles and areas between 3D triangles and their corresponding 2D counterparts. In practice, we adopt the same formulations of DDL and TDL as proposed in (Zhao et al., 2025), and refer readers to (Sheffer et al., 2007b; Hormann et al., 2007; Floater & Hormann, 2005) for further details on distortion metrics and differential geometry concepts.

### A.3 ADDITIONAL EXPERIMENTS AND RESULTS

We show expanded results, ablation studies, and discussions from the main paper featuring more surface parameterization and more viewpoints of different 3D objects in Figs. 6- 15, and Tables. 5-9.

**Ablation studies on different UV packing (aggregator) methods.** To further improve our semantic-aware UV parameterization pipeline's output, a simple enhancement is to post-process the UV charts by jointly optimizing two key UV metrics: texel density and UV space utilization. Higher texel density improves the quality of the projected 3D texture, and maintaining consistent texel density across charts ensures uniform visual fidelity. At the same time, efficient space utilization maximizes the use of available texture memory. Hence, to analyze the effect of more advanced UV-chart packing (aggregator) strategies (different from the one proposed in Sec. 3.3), we perform an ablation study and evaluate several UV-packing techniques from three widely used commercial tools including, Autodesk Maya (Autodesk, Inc., 2025), Houdini (Toronto-based SideFX, 2025), and Blender (Blender Foundation, 2025). We use a 3D cow mesh shown in Fig. 6 to evaluate texel density and UV-space utilization of UV parameterizations packed using different UV packing strategies from the three commercial tools. Specifically, we apply our semantic-aware UV parameterization framework as follows: (i) the 3D cow is first partitioned by PartField (Liu et al., 2025) into seven semantic 3D parts, (ii) then the geometry-preserving neural backbone is applied separately to each part to learn UV parameterizations for each 3D part, (iii) instead of using our default UV packing (presented in Sec. 3.3), we apply different UV packing strategies from the three commercial tools, and (iv) lastly, for each packing strategy, we compute two UV metrics, namely texel density and UV-space utilization, for the UV parameterizations resulting from the three different UV packing methods. Table 5 reports the resulting values for both metrics to illustrate how each tool performs on a 3D cow mesh example. Also, in Fig. 6, we qualitatively compare the three commercial tools, applying their automatic packing and layout features to the same unwrapped mesh of 3D cow.

**Ablation studies on different partitioning strategies (ShDF vs. SAMesh vs. PartField).** We replace our default Shape Diameter Function (ShDF) partitioner (Shapira et al., 2008) with SAMesh (Tang et al., 2024) and PartField (Liu et al., 2025) while keeping the remainder of the partition-and-parameterize (semantic-aware UV parameterization) pipeline identical. Then, we train each pipeline separately on input 3D meshes and measure and report angle-preservation (conformality) and area-preservation (equiareality) properties in the generated UV parameterization. Figs. 13- 14 and Table 6 shows the qualitative and quantitative results of these experiments, respectively. To interpret the numbers in the table, we point out two points: (i) There is no obvious or general relationship between the choice of segmentation algorithm and the geometry-preservation metrics (conformality and equiareality) of the final UV parameterization. This is due to the fact that existing 3D segmentation methods were developed to produce semantically meaningful parts, not to optimize downstream UV parameterization. Consequently, they do not incorporate UV-specific information. Therefore,

Table 5: Quantitative ablations of different UV-packing strategies from three widely used commercial tools: Autodesk Maya, Blender, and Houdini. We use a 3D cow mesh shown in Fig. 6 to evaluate texel density and UV-space utilization of UV parameterizations packed using different UV packing strategies from the three commercial tools. Specifically, after applying our semantic-aware UV parameterization framework, we obtain UV parameterizations for each segmented 3D part, resulting in seven UV charts. Then, instead of using our default UV packing (presented in Sec. 3.3), we apply different UV packing strategies from the three commercial tools. Lastly, for each packing strategy, we compute two UV metrics, namely texel density and UV-space utilization, for the UV parameterizations resulting from the three different UV packing methods. Higher texel density improves the quality of the projected 3D texture, and maintaining consistent texel density across charts ensures uniform visual fidelity. At the same time, efficient space utilization maximizes the use of available texture memory.

| Method | Texel Density | | | | | | | | UV Space Utilization |
|---|---|---|---|---|---|---|---|---|---|
| | Chart1 | Chart2 | Chart3 | Chart4 | Chart5 | Chart6 | Chart7 | Avg | |
| Maya (Autodesk, Inc., 2025) | 160.62 | 150.41 | 150.41 | 150.41 | 156.51 | 150.41 | 150.41 | 152.74 | 71% |
| Blender (Blender Foundation, 2025) | 151.66 | 142.02 | 142.02 | 142.02 | 147.78 | 142.02 | 142.02 | 144.22 | 57% |
| Houdini (Toronto-based SideFX, 2025) | 157.08 | 147.09 | 147.09 | 147.09 | 153.06 | 147.09 | 147.10 | 149.37 | 62% |

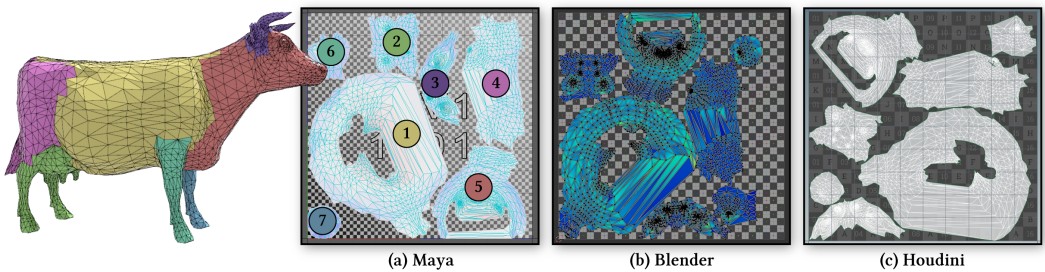

(a) Maya        (b) Blender        (c) Houdini

Figure 6: Qualitative ablations of different UV-packing strategies. To further improve our semantic-aware UV parameterization pipeline's output, we present alternative more advanced UV packing results (different from the one proposed in Sec. 3.3) obtained from three widely used commercial tools: (a) Autodesk Maya, (b) Blender, and (c) Houdini. The left-most column shows the 3D cow mesh segmented by PartField (Liu et al., 2025). In the Autodesk Maya result (a), each UV chart corresponding to a semantic 3D part is colored to match its counterpart on the 3D mesh. The same applies to the Blender and Houdini results. Specifically, the 3D cow is first partitioned by PartField (Liu et al., 2025) into seven semantic 3D parts. Then, the geometry-preserving neural backbone (Sec. 3.2) is applied separately to each part to learn UV parameterizations for each 3D part leading to seven UV charts. Then, instead of using our default UV packing (presented in Sec. 3.3), we apply automatic packing strategies layout features of the three commercial tools to the unwrapped mesh of 3D cow.

from the numbers in Table 6, we cannot draw a unified, causal conclusion connecting segmentation method type to final UV geometry properties. Having that said, we include this segmentation ablation and Table 6 intentionally to inform practitioners that different partitioning methods, while varying in semantic quality, can also influence geometry-preservation in the resulting UVs. Reporting these results shows the important trade-off between semantic-awareness and geometric fidelity. Therefore, practitioners should try different segmentation approaches and choose the one that best fits their application needs. (ii) Of course, state-of-the-art, learning-based segmentation methods can produce more semantically coherent partitions than ShDF (Shapira et al., 2008) (as shown by works such as SAMesh (Tang et al., 2024)). Therefore, any improvements in that literature can directly improve the semantic-awareness of our pipeline because segmentation is the first stage of our semantic-aware parameterization (Fig.2). In other words, a better 3D partitioner is an immediate plug-in improvement for our method.

**Ablation studies on geometry-preservation vs. semantic / visibility objectives.** Optimizing multiple geometry-preservation metrics simultaneously (conformality, equiareality, stretch, bijectivity, etc.) is known to require trade-offs. Specifically, achieving the best possible value for every metric at once is generally impractical. Introducing higher-level objectives (semantic- and visibility-awareness), that we propose in this work, increases this tension and therefore can further degrade

some geometry metrics. To quantify these effects, we includ an ablation in Table. 7 that compares our proposed pipelines to the baseline (labelled "Base Neural UV Arch" in Table. 7, i.e., FlexPara) which is state-of-the-art for geometry preservation. We compares three variants of our pipeline to the baseline: (i) Semantic-Aware Param, the pipeline of Sec. 3.3 (Fig. 2), which first partitions the mesh into semantic parts and then learns per-part parameterizations; (ii) Visibility-Aware Param, the pipeline of Sec. 3.4 (Algorithm. 1), which produces a global atlas while steering seams toward less-visible regions; and (iii) Visibility+Semantic-Aware Param, the combined pipeline that applies both sets of objectives. Three observations explain the quantitative trends: (i) when comparing conformality and equiareality, the Semantic-Aware and the combined Visibility+Semantic-Aware pipelines lose little equiareality relative to the baseline and even show slight improvement in conformality in some cases. Intuitively, this is because the semantic-aware stage first partitions the mesh into multiple semantically meaningful subparts. Intuitively, learning a geometry-preserving parameterization on smaller, simpler patches is easier. In other words, the network can better satisfy geometry-preservation while also enforcing semantic consistency. Therefore, semantic partitioning provides a natural way to reduce distortion without heavily sacrificing geometry metrics. (ii) By contrast, the visibility-aware pipeline does not partition the mesh. Instead, it steers seams to less-visible (more occluded) regions while producing a global atlas (a single large chart). Unwrapping a whole 3D mesh into a single UV chart to while simultaneously optimizing visibility and geometry is significantly harder. Consequently, the Visibility-Aware pipeline shows a larger drop in equiareality compared to the baseline and compared to the semantic-aware pipelines. (iii) The combined Visibility+Semantic-Aware pipeline aims to obtain the best of both worlds: it first partitions the mesh into smaller parts to retain geometry preservation, and then steers seams toward less-visible regions within those parts. This combined approach yields a more favorable trade-off: it loses far less equiareality (0.6369) than the pure visibility-aware pipeline (0.6093) while still improving seam visibility and semantic consistency.

**Analysis on training and inference computation times.** To provide users and practitioners with some insights about the computation time needed for the proposed method in this study, we report both inference and training time of the proposed method in Table 3, Table 1, Table 8, and Table 9, and compare them with the computation time of the state-of-the-art method (at the time of writing this manuscript), FlexPara (Zhao et al., 2025).

- **Inference time.** (i) Visibility-aware pipeline: although training is performed per-mesh (as is common in this literature: Nuvo (Srinivasan et al., 2024), FlexPara (Zhao et al., 2025), OptCuts (Li et al., 2018)), the visibility objective is applied only during training (not the inference stage). Specifically, at inference the trained model uses the same neural architecture as FlexPara (Zhao et al., 2025) and therefore incurs no additional runtime overhead. This is why the inference times for our Visibility-Aware Param and FlexPara are the same in Table 1 ($\approx$2 seconds). (ii) Semantic-aware pipeline: the semantic-aware pipeline does introduce extra inference cost because it performs three stages at test time: (a) 3D semantic partitioning, (b) per-part parameterization (running the backbone once per part), and (c) aggregation/UV packing. These additional steps (segmentation, multiple backbone invocations, and packing) explain the larger inference time reported in Table 2 and for the combined pipeline (semantic+visibility-aware) in Table 1. A simple and effective mitigation is to parallelize the per-part backbone calls by using multi-threading during the inference, which would significantly reduce inference time.

- **Training time.** (i) Semantic-aware pipeline (Table 8): counterintuitively, our semantic-aware pipeline achieves shorter training time than FlexPara (Zhao et al., 2025) across different 3D meshes. The reason is that the mesh is partitioned into many smaller submeshes: although our semantic-aware pipeline calls the backbone multiple times (once per semantic part), which might suggest longer training, this is compensated by the fact that the 3D segmented parts are much smaller than the original input mesh used by FlexPara. Hence, training on many small submeshes is faster than training a single model on a large, complex mesh with many vertices and faces. This empirical observation is consistent with our careful time measurement of each stage in the semantic-aware pipeline which is: learning UV parameterization for small 3D parts is much faster than doing the same for large meshes. More importantly, the reduction in per-part training cost outweighs the multiplicative overhead of multiple backbone runs, which results in shorter overall training time for the semantic-aware pipeline in our experiments. This is why Table 8 shows shorter train-

ing times for our semantic-aware pipeline compared to FlexPara in our experiments. (ii) Visibility-aware pipeline (Table 9): The visibility-aware pipeline trains slower than Flex-Para (Zhao et al., 2025) across different meshes. This is expected because, compared to the base neural backbone (i.e., FlexPara), our visibility-aware pipeline introduces per-iteration overheads: (a) per-vertex ambient occlusion (AO) computation, (b) candidate UV generation via the backbone, (c) seam/boundary extraction in UV space, and (d) AO-weighted loss computation. The most expensive stage is seam extraction which accounts for approximately 90% of the additional training time. Specifically, for each vertex we inspect its 3–5 neighbors, compare mapped UV positions, and evaluate Eq. 5 to detect UV discontinuities. This neighbor-checking and distance computation scales with vertex count and dominates the additional training cost ( 90% of the overhead), making visibility-aware training significantly heavier for complex meshes.

- **Potential attempts to overcome computation burdens.** To mitigate the computational burdens discussed above, we propose two practical solutions: (i) parallelizing per-part backbone calls in semantic-aware pipeline. In our current implementation, the neural backbone is executed sequentially for each semantic part. Switching to a multi-threaded or multi-GPU implementation to run per-part backbone calls in parallel would significantly reduce both inference and training times. (ii) Optimizing seam detection in visibility-aware pipeline. The seam-detection stage is the main training-time bottleneck. Algorithmic and engineering improvements, e.g., spatial hashing, cached neighborhood lookups, and GPU-accelerated neighbor/distance checks, can significantly reduce this overhead. We note these optimizations as straightforward improvements and sensible directions for future work.

Table 6: Ablation studies on the proposed partitioning strategy (ShDF vs. SAMesh vs. PartField). We replace our default Shape Diameter Function (ShDF) partitioner (Shapira et al., 2008) with SAMesh (Tang et al., 2024) and PartField (Liu et al., 2025) while keeping the remainder of the *partition-and-parameterize* pipeline identical. Quantitatively, using ShDF in our pipeline yields UV parameterizations that are more conformal (better angle preservation) and more equiareal (better area preservation) than those produced when using SAMesh.

| Method | Conformality ↑ | Equiareality ↑ |
|---|---|---|
| Ours - Semantic-Aware Param (PartField) | 0.8999 | **0.6921** |
| Ours - Semantic-Aware Param (SAMesh) | 0.8694 | 0.6270 |
| Ours - Semantic-Aware Param (ShDF) | **0.9123** | 0.6707 |

Table 7: Ablation: the effect of semantic and visibility objectives on geometry-preserving metrics. Reported are mean Conformality and Equiareality (higher is better). The Semantic-Aware and the combined Visibility+Semantic-Aware pipelines lose little equiareality relative to the baseline and even show slight improvement in conformality in some cases. Intuitively, this is because the semantic-aware stage first partitions the mesh into multiple semantically meaningful subparts and so the network can better satisfy geometry-preservation while also enforcing semantic consistency. However, the visibility-aware pipeline does not partition the mesh. Instead, it steers seams to less-visible (more occluded) regions while producing one global atlas (a single large chart). Unwrapping and optimizing a single global chart to satisfy both geometry-preservation and visibility objectives is significantly harder. Consequently, the Visibility-Aware pipeline shows a larger drop in equiareality compared to the baseline. The combined Visibility+Semantic-Aware pipeline aims to obtain the best of both worlds: (i) partition the mesh into smaller parts to retain geometry preservation, and (ii) steer seams toward less-visible regions within those parts. This combined approach yields a favorable trade-off: it loses far less equiareality (0.6369) than the pure visibility-aware pipeline (0.6093) while still improving seam visibility and semantic consistency.

| Method | Conformality ↑ | Equiareality ↑ |
|---|---|---|
| Base Neural UV Arch. | 0.9097 | **0.6759** |
| Ours - Semantic-Aware Param | 0.9123 | 0.6707 |
| Ours - Visibility-Aware Param | **0.9175** | 0.6093 |
| Ours - Visibility+Semantic-Aware Param | 0.9153 | 0.6369 |

Table 8: Comparison of training time between our proposed semantic-aware parameterization pipeline and FlexPara's multi-chart parameterization (Zhao et al., 2025). Total times are shown outside the parentheses, and per-iteration times are shown inside.

| Method | Training Time (Seconds) | | | |
|---|---|---|---|---|
| | Rabbit | Potion | Squidward House | **Average** |
| FlexPara (multi-Chart) (Zhao et al., 2025) | 900 (0.0900) | 16377 (1.6377) | 13784 (1.3784) | 10353 (1.0353) |
| Ours - Semantic-Aware Param | 676 (0.1352) | 666 (0.1332) | 602 (0.1205) | **648 (0.1296)** |

Table 9: Comparison of training time between our proposed visibility-aware parameterization pipeline and FlexPara's global (single-chart) parameterization (Zhao et al., 2025). Total times are shown outside the parentheses, and per-iteration times are shown inside.

| Method | Training Time (Seconds) | | | | | |
|---|---|---|---|---|---|---|
| | Duckie | Fish | FlipFlop | Rabbit | Spot | **Average** |
| FlexPara (single-Chart) (Zhao et al., 2025) | 1306 (0.1306) | 1323 (0.1323) | 1275 (0.1275) | 1293 (0.1293) | 1230 (0.1230) | **1285 (0.1285)** |
| Ours - Visibility-Aware Param | 36642 (3.6642) | 47460 (4.7360) | 16864 (3.3729) | 35232 (2.3488) | 19605 (1.9605) | 31160 (3.1160) |

Table 10: To evaluate semantic- and visibility-awareness of the proposed method, we conducted a user-study with 70 general participants (including graduate students with computer science and engineering backgrounds) performing 11 comparisons between textured 3D shapes and UV parameterizations produced by our method and baselines. We report the percentage of general participant preferences for each method. Our proposed method is strongly preferred by the general users over the baselines.

| Method | General User Preference Percentage |
|---|---|
| **Visibility-Awareness Evaluation** | |
| OptCuts (Li et al., 2018) | 1.43 % |
| FlexPara (single-Chart) (Zhao et al., 2025) | 7.14 % |
| Ours - Visibility-Aware Param | **91.42 %** |
| **Semantic-Awareness Evaluation** | |
| xatlas (jpcy, 2025) | 4.28 % |
| Blender (Blender Foundation, 2025) | 4.28 % |
| Autodesk Maya (Autodesk, Inc., 2025) | 10 % |
| FlexPara (multi-Chart) (Zhao et al., 2025) | 7.14 % |
| Ours - Semantic-Aware Param | **74.29 %** |

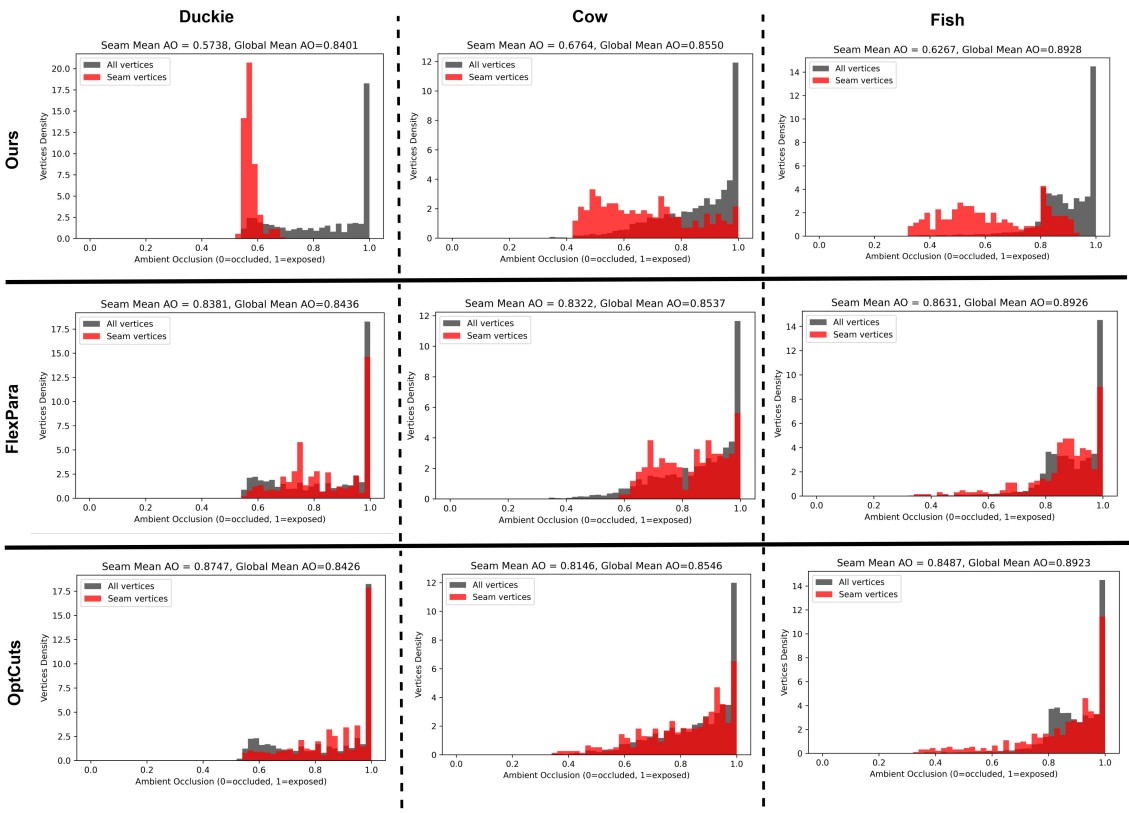

Figure 7: Quantitative distribution of seam-vertex ambient occlusion (AO) for three meshes parameterized with our method, FlexPara (Zhao et al., 2025), and OptCuts (Li et al., 2018). Red bars indicate AO distributions of seam vertices, while black bars represent all vertices. The $y$-axis denotes vertex density and the $x$-axis AO values. Our method shifts seam-vertex AO distributions toward lower values (left side), showing that the visibility-aware pipeline relocates seams to less visible (more occluded) regions, unlike FlexPara and OptCuts.

## B    THE USE OF LARGE LANGUAGE MODELS

We used a large language model (LLM) to assist with editing, refining, and polishing the manuscript text (for example: abstract and captions). The LLM was used only for language refinement, improving grammar, wording, concision, and overall readability, and did not contribute to the technical content, experiments, results, or scientific claims. All suggestions generated by LLM were manually and carefully reviewed and, where appropriate, revised by the authors. The authors have full responsibility for the final text and for the accuracy of the work.

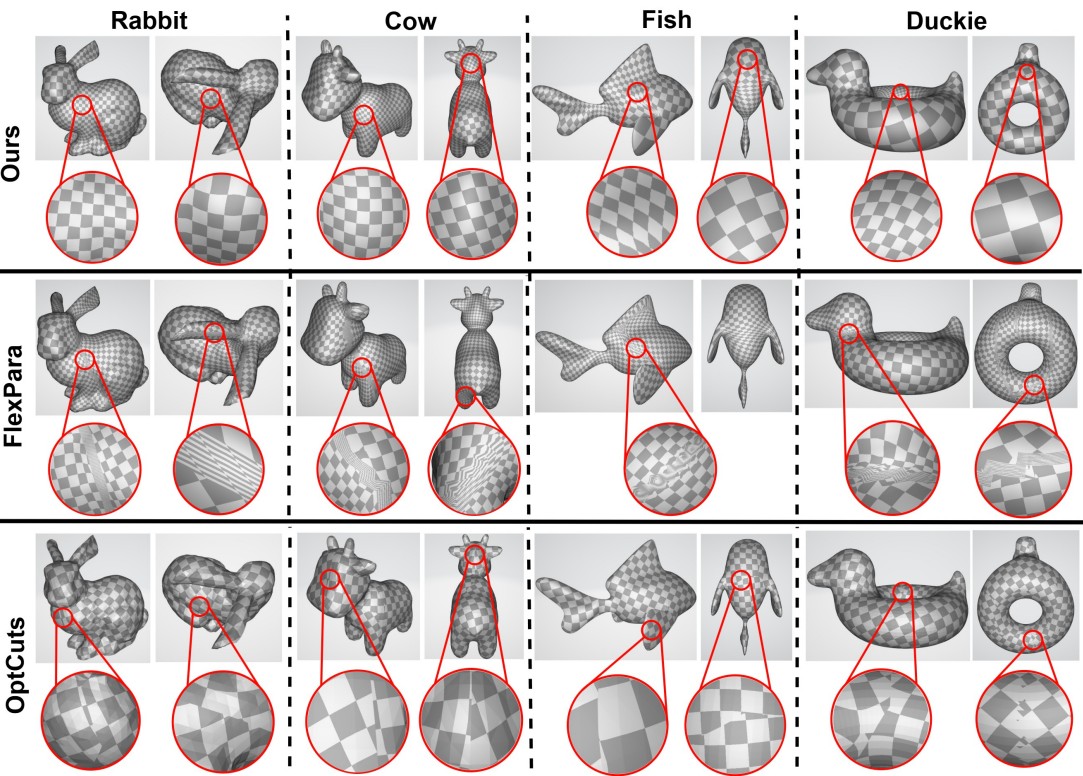

Figure 8: Extended version of Fig. 5 in the main paper. Checkerboard texturing comparison using UV parameterizations produced by our visibility-aware method, FlexPara (Zhao et al., 2025), and OptCuts (Li et al., 2018). Each row shows rendered views of different meshes textured with a checkerboard and a magnified inset of a visually important region near seams (red circles). Because our method steers seams toward occluded regions, the checkerboard pattern appears substantially more continuous from typical camera viewpoints. By contrast, baselines exhibit visible seam artifacts in the zoomed-in insets.

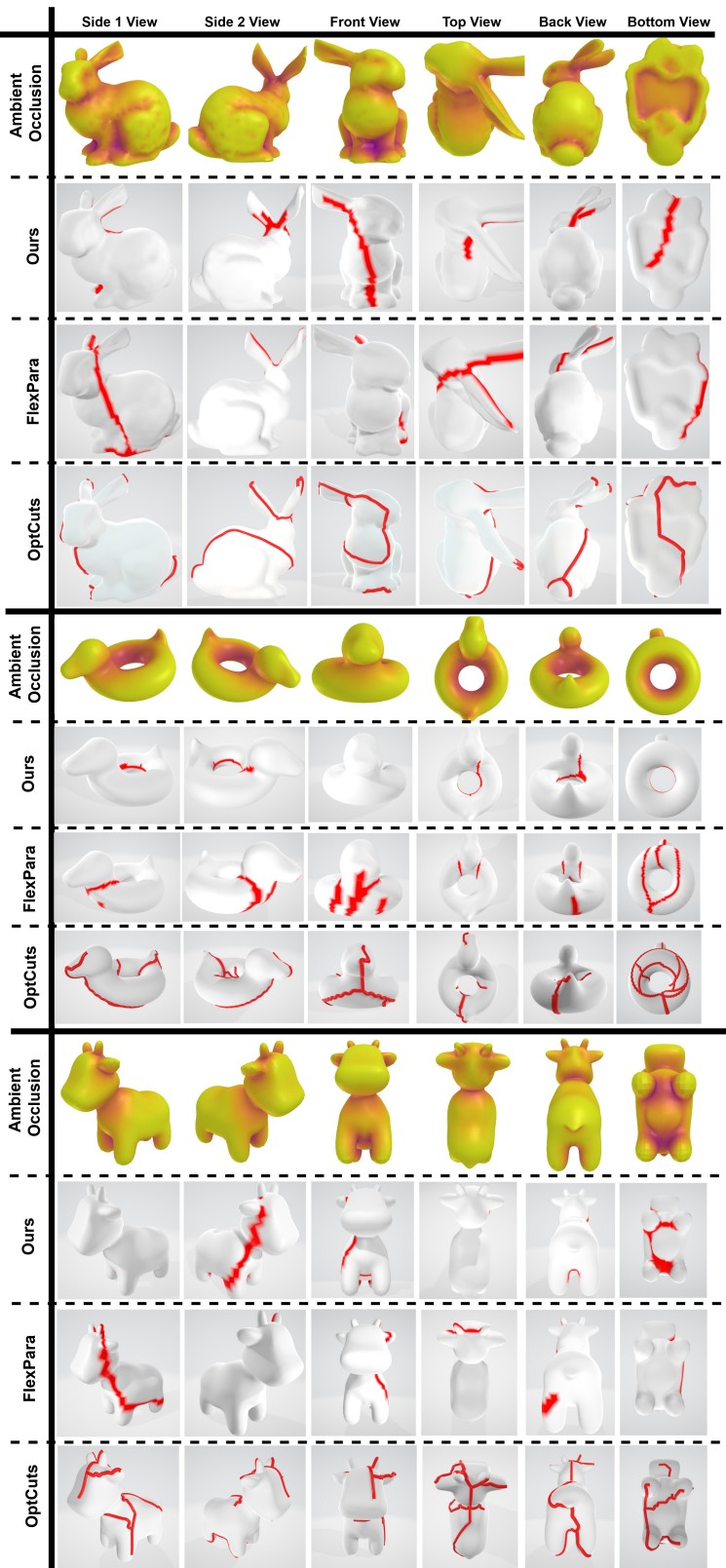

Figure 9: Extended version of Fig. 4 in the main paper. Qualitative results for visibility-aware seam placement and UV parameterization on three representative meshes. For each mesh, the top row shows per-vertex ambient occlusion (yellow = exposed, purple = occluded). Beneath are the visualizations of cutting seams (red) from our method, FlexPara (Zhao et al., 2025), and OptCuts (Li et al., 2018) (top to bottom). Our method places a larger fraction of seam geometry in less-exposed regions, reducing the likelihood of visible seam artifacts under typical viewpoints.

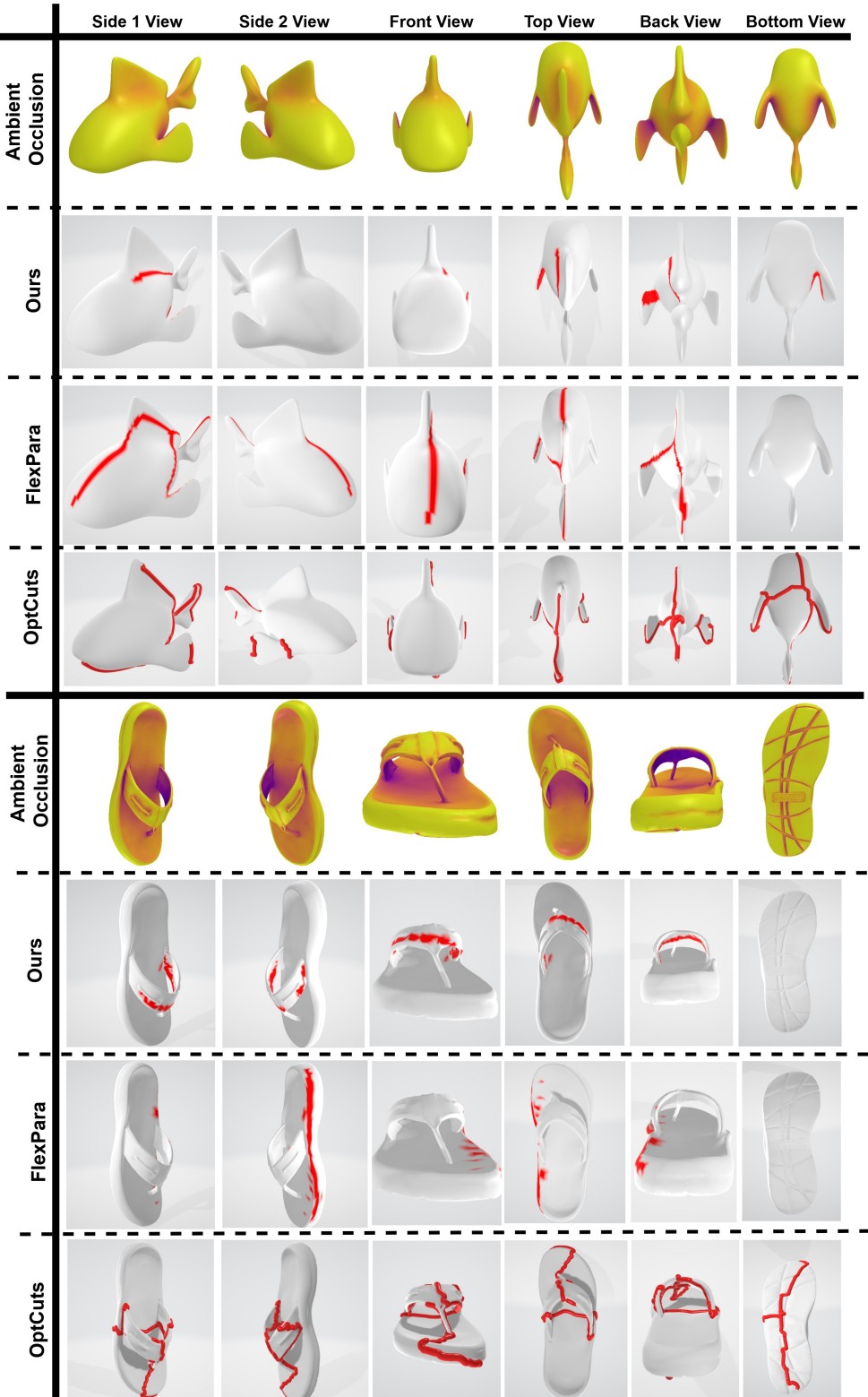

Figure 10: Extended version of Fig. 4 in the main paper. Qualitative results for visibility-aware seam placement and UV parameterization on three representative meshes. For each mesh, the top row shows per-vertex ambient occlusion (yellow = exposed, purple = occluded). Beneath are the visualizations of cutting seams (red) from our method, FlexPara (Zhao et al., 2025), and OptCuts (Li et al., 2018) (top to bottom). Our method places a larger fraction of seam geometry in less-exposed regions, reducing the likelihood of visible seam artifacts under typical viewpoints.

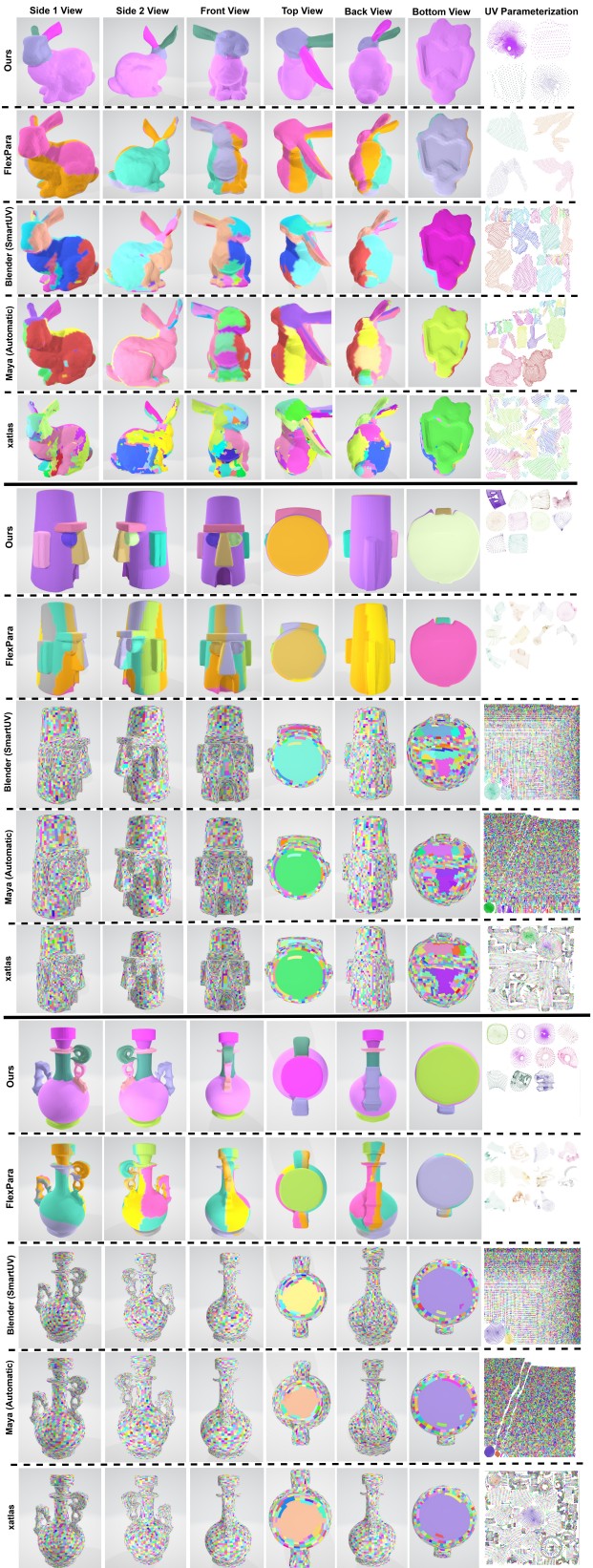

Figure 11: Extended version of Fig. 3 in the main paper. Qualitative results of the proposed semantic-aware UV parameterization method on a Rabbit mesh. For each method, we show the rendered 3D objects from multiple viewpoints, with the corresponding UV atlas in the rightmost column. As shown, our method produces UV charts that are align more semantically with the mesh's 3D semantic parts, unlike the baselines.

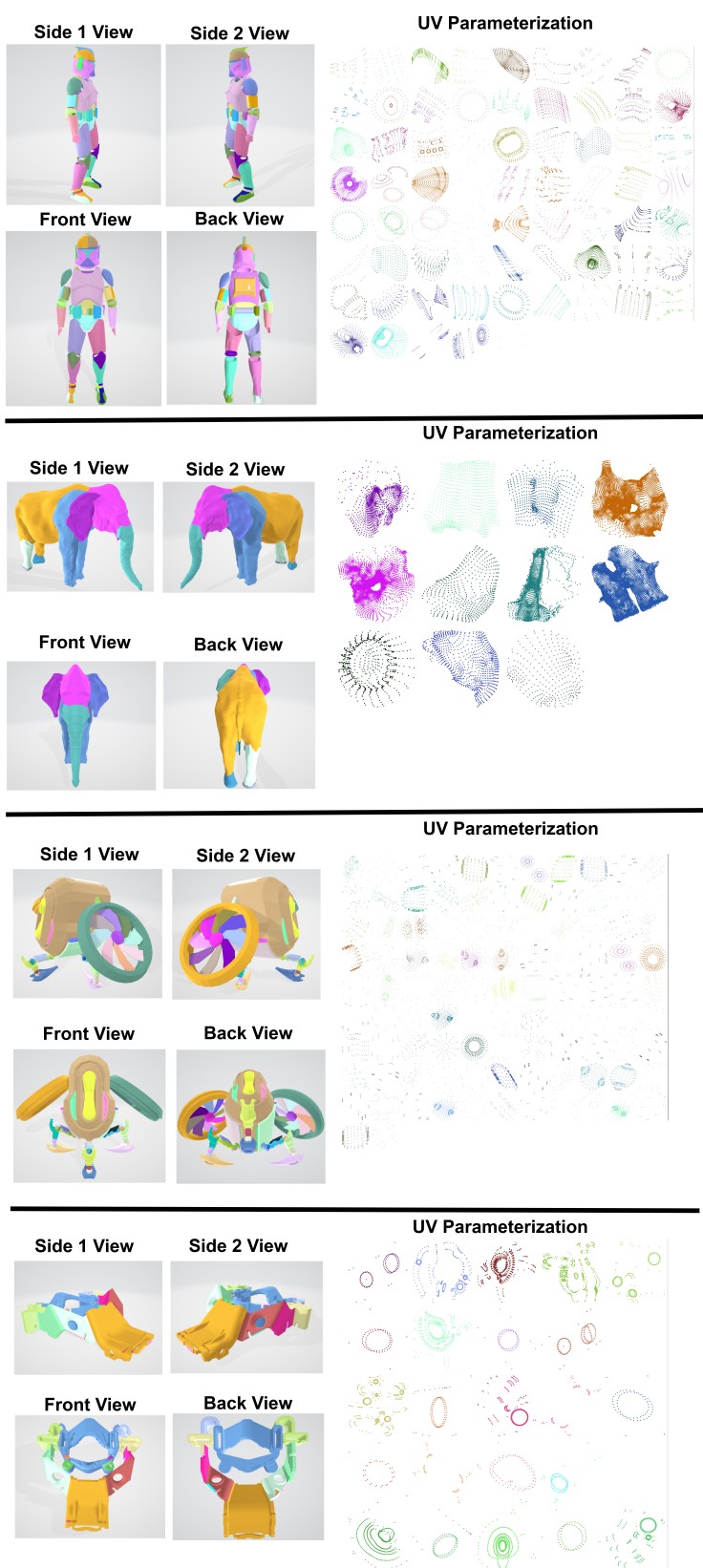

Figure 12: Qualitative results of the proposed semantic-aware UV parameterization method on complex mesh objects with high number of triangle faces. For each method, we show the rendered 3D objects from multiple viewpoints, with the corresponding UV atlas in the rightmost column. As shown, our method is able to produce UV charts, for complex meshes, that are align more semantically with the mesh's 3D semantic parts

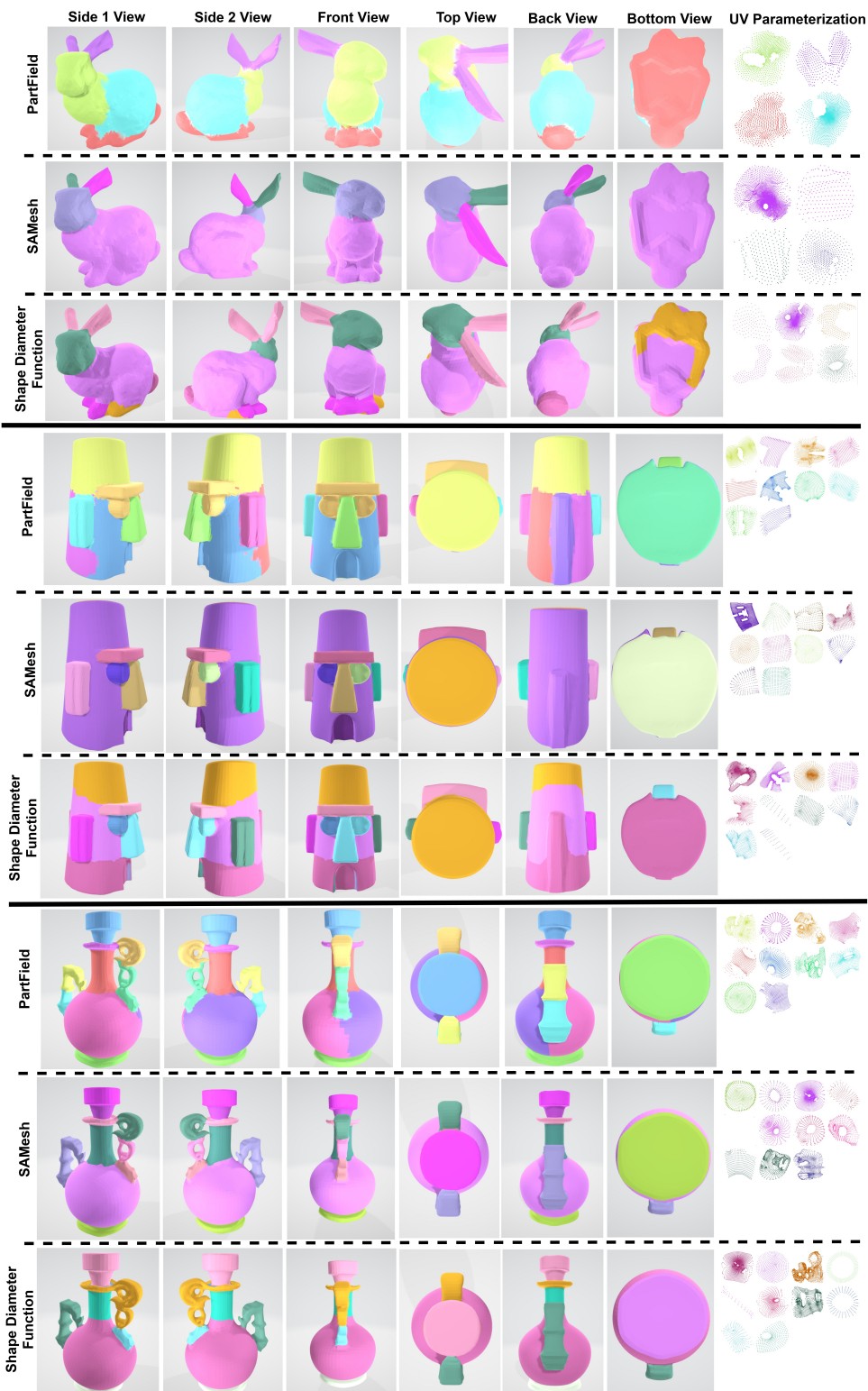

Figure 13: Ablation studies on the proposed partitioning strategy (ShDF vs. SAMesh vs. PartField) of simple meshes. We replace our default Shape Diameter Function (ShDF) partitioner (Shapira et al., 2008) with SAMesh (Tang et al., 2024) and PartField (Liu et al., 2025) while keeping the remainder of the *partition-and-parameterize* pipeline identical. Qualitatively all three partitioners can produce part decompositions that align with geometric structure across simple meshes, resulting compact, low-distortion per-part UV charts. However, it seems that PartField produces more accurate and more detailed semantic segmentations than ShDF and SAMesh in some cases (e.g., the feet and the tail are correctly detected, whereas they are not detected properly in SAMesh).

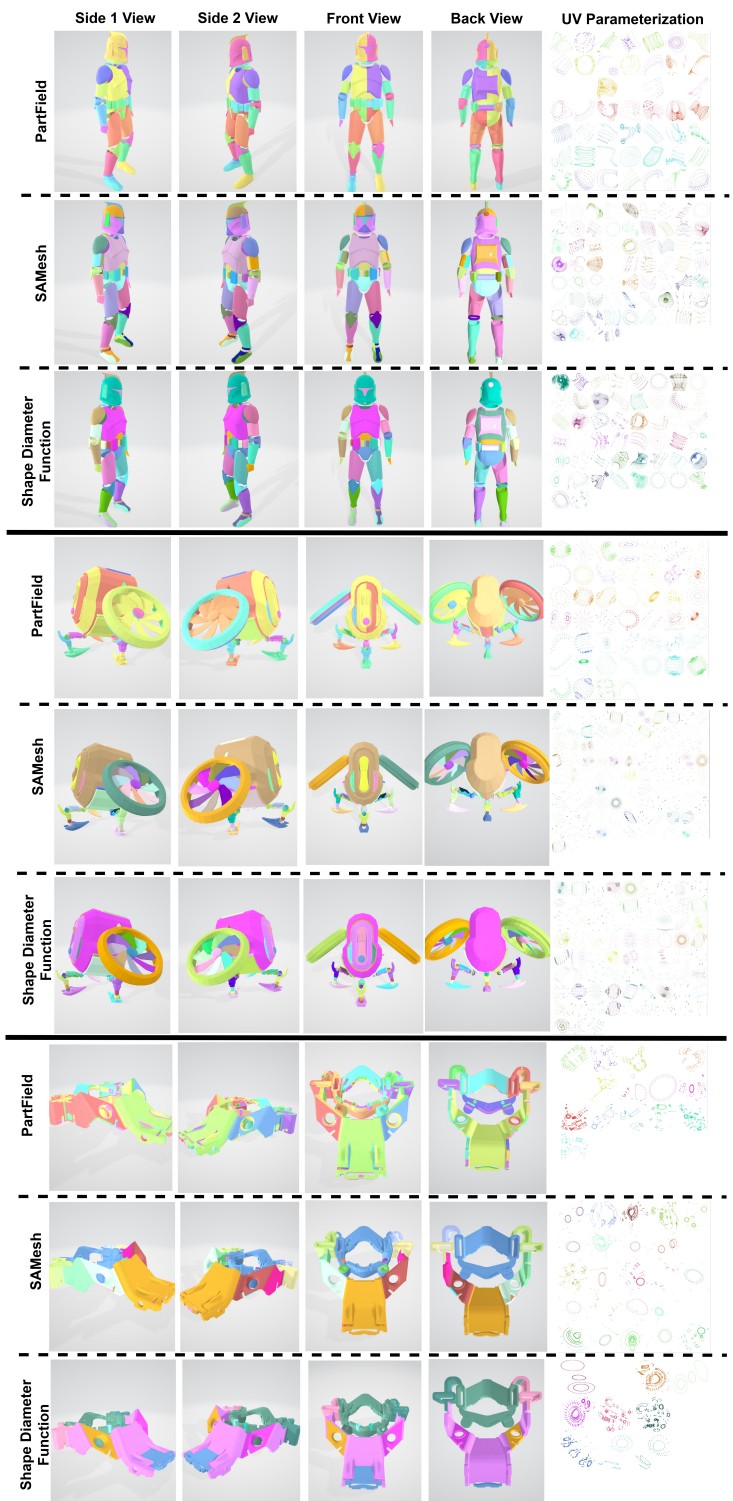

Figure 14: Ablation studies on the proposed partitioning strategy (ShDF vs. SAMesh vs. PartField) of complex meshes with high number of triangle faces. We replace our default Shape Diameter Function (ShDF) partitioner (Shapira et al., 2008) with SAMesh (Tang et al., 2024) and PartField (Liu et al., 2025) while keeping the remainder of the *partition-and-parameterize* pipeline identical. Qualitatively all partitioners produce part decompositions that align with geometric structure across complex models. However, PartField appears to produce more accurate and more compact semantic segmentations than ShDF and SAMesh in some cases. For example, PartField correctly groups the entire set of fan blades into a single segmented part, whereas SAMesh and ShDF split each blade into a separate part. This is not completely wrong, but it results in too many small charts after per-part UV parameterization, which may be undesirable for some downstream tasks.

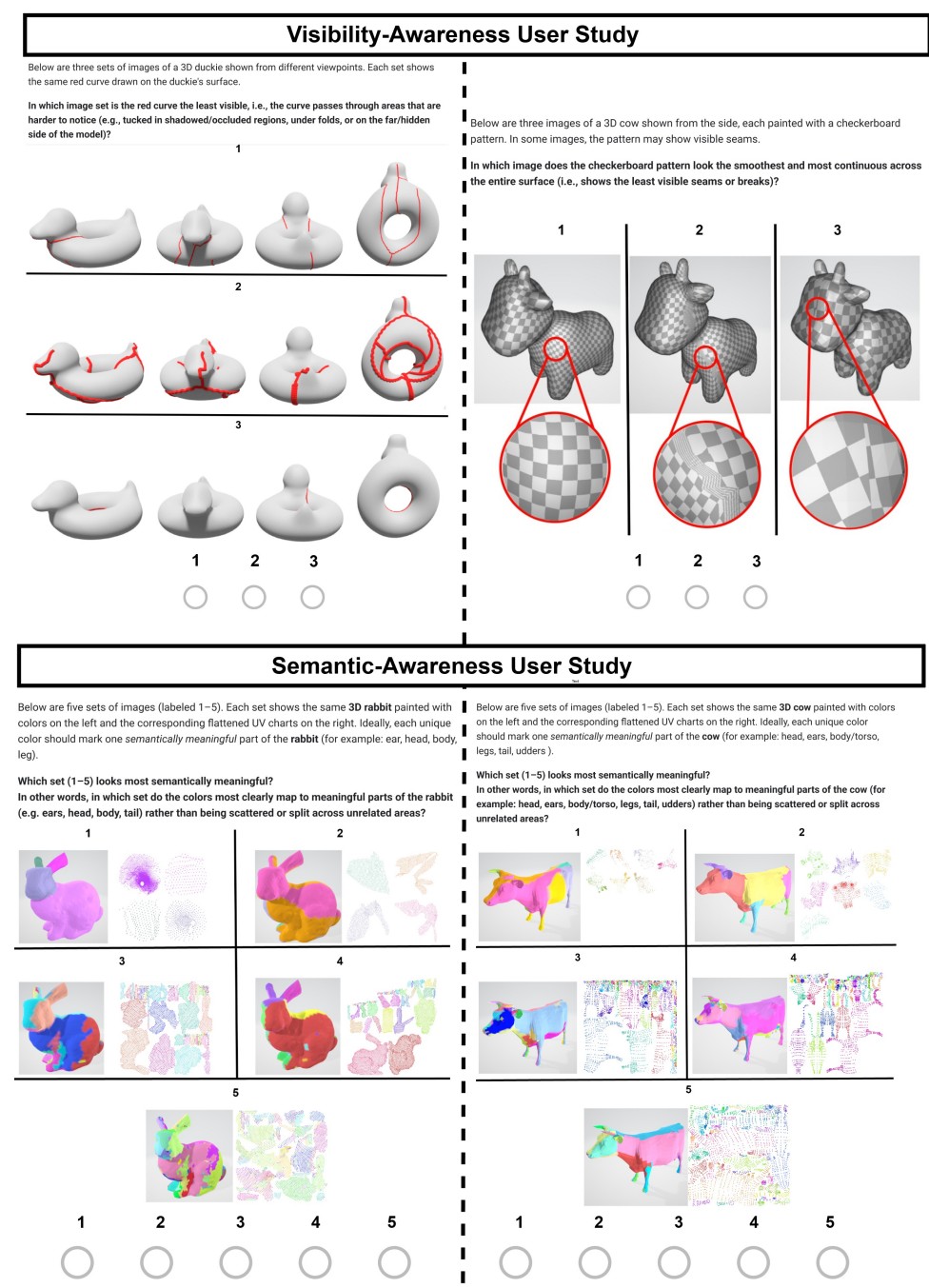

Figure 15: Examples of four of the eleven questions in our user-study questionnaire. To evaluate semantic- and visibility-awareness of the proposed method, we conducted a user-preference study with 115 participants. We distributed the questionnaire to two groups: experts and general participants. Of the 115 participants, 70 are general participants (including graduate students with computer science and engineering backgrounds) and 45 are experts: 31 software engineers, 3 project managers, 2 product owners, 5 UV/layout artists, and 4 modeling artists, all working in the Media and Entertainment industry for film and games. Each participant completed 11 comparisons between textured 3D shapes and UV parameterizations produced by our method, FlexPara (Zhao et al., 2025), OptCuts (Li et al., 2018), Autodesk Maya (Autodesk, Inc., 2025), Blender (Blender Foundation, 2025), and xatlas (jpcy, 2025). For each comparison, participants were rated each result according to three visual criteria: (i) texture-pattern smoothness/continuity, (ii) semantic alignment (how well colors correspond to meaningful parts), and (iii) seam visibility (how well seams are placed in occluded/less-obvious regions). The semantic-aware and visibility-aware evaluations comprised 5 and 6 questions, respectively, for a total of 11 comparisons per participant. Tables 2 and 10 report the percentage of expert and general participant preferences for each method, respectively. As the tables show, our proposed method is strongly preferred over the baselines by both expert industry-level users and general users.

