# OpenReview forum: "Unsupervised Representation Learning for 3D Mesh Parameterization with Semantic and Visibility Objectives"
_ICLR.cc/2026/Conference — ICLR 2026 Poster_

### Official Review · Reviewer_Gh6M · 2025-10-26

**Soundness:** 3
**Presentation:** 3
**Contribution:** 3
**Rating:** 6
**Confidence:** 4

**Summary:**

The proposed method builds upon previous work, such as FlexPara and Nuvo, which address UV unwrapping. These methods propose a neural network optimization-based unwrapping approach with mapping cycles. The neural networks employed are neural fields similar to those utilized in NeRF. Compared to previous methods, the proposed approach incorporates more meaningful cut control based on visibility and semantics. This significantly improves the cut positions and generates more meaningful islands. For semantics, the authors propose using the shape diameter function (ShDF), while also demonstrating a version employing a semantic network. Regarding visibility-based cutting, they propose utilizing the ambient occlusion (AO) as a measurement. Both losses can be utilized jointly during the optimization process.

**Strengths:**

- The utilization of AO as a measurement to identify concealed cut areas is a novel approach. The authors’ integration of this information (soft-selection) into the optimization process is noteworthy.
- Parallel to the AO integration, the incorporation of part segmentation employing ShDF and Gaussian mixture models to learn the soft assignment is also interesting.
- While the SAMesh version appears more preferable to me from the supplementary materials, the application of ShDF yields remarkably convincing results, considering the simplicity of the technique.
- The authors comprehensively evaluate their method employing various metrics, demonstrating the enhanced part segmentation.

**Weaknesses:**

- The current level of UV unwrapping is still not on artist-levels, but it does make significant improvements over previous works.
- The chart layout is intentionally straightforward, but I could envision easy modifications to maintain a similar texture density.
- Overall, the method only adds a small delta on-top of the existing methods (Semantic and Visibility-based cutting).

**Questions:**

- Could you please provide the training duration per mesh? n Table 2, there is an inconsistency in the inference times. Since training is performed per asset, it should be factored in. FlexPara estimates times of approximately 400 seconds for a global unwrap and 1300 seconds for a multi-chart, which is comparable to Nuvo’s speed. I would anticipate that the method would require slightly longer training times.

---

> ### Author Response · Authors · 2025-11-21
> **Response to Reviewer Gh6M - Weakness 1**
>
> We thank the reviewer for recognizing the importance and impact of this work.
>
> We agree that there remains room to reach the industry-level UV quality that artists expect. Nevertheless, to the best of our knowledge, our work is the first learning-based approach to specifically target this gap by integrating semantic- and visibility-aware objectives into the UV-parameterization pipeline. We view this contribution as an important step toward automating high-quality, artist-friendly UV workflows and hope it opens a promising direction for future research and practical refinement.

---

> ### Author Response · Authors · 2025-11-21
> **Response to Reviewer Gh6M - Weakness 2**
>
> We thank reviewer for this insightful remark. Our current post-processing stage applies only basic refinements to the final UVs, but incorporating texel-density analysis and chart rescaling to ensure consistent texture density across all charts is indeed a valuable enhancement. We initially assumed that end users and artists would prefer handling these finer adjustments in their preferred tools. On the other hand, we agree that offering a more consistent and robust starting point would strengthen our pipeline, and we will incorporate this consideration moving forward.
>
> As reviewer vZwh also highlighted, jointly accounting for texel density and space utilization is crucial to avoid suboptimal use of the normalized UV domain, an important factor in production workflows, particularly for real-time applications. These thoughtful observations also suggest an interesting direction for future work: integrating new metrics into the loss function to further improve Unwrap-Net, in the spirit of the approach proposed by the authors of FlexPara.
>
> Therefore, as reviewers vZwh and fuqi also noted regarding our UV aggregator, to further improve our semantic-aware UV parameterization pipeline's output, a simple enhancement is to post-process the UV charts by jointly optimizing two key UV metrics: texel density and UV space utilization. Higher texel density improves the quality of the projected 3D texture, and maintaining consistent texel density across charts ensures uniform visual fidelity. At the same time, efficient space utilization maximizes the use of available texture memory.
>
> Hence, to analyze the effect of more advanced UV-chart packing (aggregator) strategies (different from the one proposed in Sec.3.3), we perform an ablation study and evaluate several UV-packing techniques from three widely used commercial tools including, Autodesk Maya, Houdini, and Blender. We use a 3D cow mesh shown in Fig.7 to evaluate texel density and UV-space utilization of UV parameterizations packed using different UV packing strategies from the three commercial tools. Specifically, we apply our semantic-aware UV parameterization framework as follows: (i) the 3D cow is first partitioned into seven semantic 3D parts, (ii) then the geometry-preserving neural backbone is applied separately to each part to learn UV parameterizations for each 3D part, (iii) instead of using our default UV packing (presented in Sec.3.3), we apply different UV packing strategies from the three commercial tools, and (iv) lastly, for each packing strategy, we compute two UV metrics, namely texel density and UV-space utilization, for the UV parameterizations resulting from the three different UV packing methods. We have now added Table 5 to the revised manuscript. This table reports the resulting values for both metrics to illustrate how each tool performs on a 3D cow mesh example. Also, we have now added Fig. 7 to the revised manuscript as a qualitative comparison of this ablation study. Fig. 7 qualitatively compares the three commercial tools, applying their automatic packing and layout features to the same unwrapped mesh of 3D cow.
>
>
> A detailed description of these strategies is provided in Appendix A.3 of the revised manuscript, and the new material is highlighted in blue.
>
>
> Finally, we emphasize the modularity of our pipeline: In the semantic-aware UV parameterization pipeline, the UV aggregator (packing) stage is interchangeable. This means that if a user prefers a different UV packing method (e.g., Autodesk Maya, Houdini, Blender, etc.), it can be plugged into our semantic-aware UV pipeline with minimal modifications.
>
>
> We appreciate the reviewer’s constructive and valid feedback. It motivated us to try multiple UV-packing strategies and compare them both qualitatively and quantitatively. These experiments demonstrate the generalizability of our semantic-aware UV-parameterization framework (it works with different packing strategies) and, importantly, provide users and practitioners with verified insights about how different packing choices affect the final UV parameterization.

---

> ### Author Response · Authors · 2025-11-21
> **Response to Reviewer Gh6M - Weakness 3**
>
> We thank the reviewer for this observation. We agree that at first glance our changes may appear incremental, but we respectfully disagree that the overall contribution is small.
>
> Specifically, the paper makes two substantive and complementary contributions:
>
> 1- A joint and unsupervised differentiable framework that simultaneously learns low-distortion UV parameterizations and directly optimizes semantic- and visibility-aware objectives. Making these objectives differentiable and training them together in a single gradient-based pipeline is, to our knowledge, new and enables end-to-end optimization that previous pipelines do not offer.
>
> 2- Two novel perceptual objectives that connect UV parameterization to downstream texture synthesis: (i) a semantic-aware loss that aligns UV charts with semantically meaningful 3D parts, and (ii) a visibility-aware loss that steers seams toward less-visible regions to reduce seam artifacts. These objectives are engineered to operate on top of an existing geometry-preserving backbone and, when combined with the backbone, yield UVs that significantly improve practical usability for texture-based tasks.
>
> Beyond the two technical contributions above, an additional contribution is the design and integration effort: we show how to combine these objectives with a state-of-the-art geometry backbone, provide ablations that isolate each component’s effect, and demonstrate empirical gains against strong open-source and commercial baselines (e.g., FlexPara, Autodesk Maya). While the per-component changes may look modest in isolation, their combination within a single, differentiable system produces meaningful qualitative and quantitative improvements for texture-aware applications, and, importantly, establishes a new, extensible direction for future work toward industry-grade automated UV tooling.
>
> We appreciate the reviewer’s scrutiny. It helped us make the novelty and impact more explicit.

---

> ### Author Response · Authors · 2025-11-21
> **Response to Reviewer Gh6M - Question 1 (part 1)**
>
> We thank the reviewer for this careful and constructive question. To remove any confusion about the reported inference times (Tables 1 and 3) and the newly added training times, we summarize the key points below and refer the reader to the newly added Tables 8–9 and Appendix A.3 for training times.
>
> ### 1- Inference time
>
> * **Visibility-aware pipeline.** Although training/optimization is performed per-mesh (as is common in this literature: Nuvo [1], FlexPara [2], OptCuts [3]), the visibility objective is applied only during training (not the inference stage). Specifically, at inference the trained model uses the same neural architecture as FlexPara and therefore incurs no additional runtime overhead. This is why the inference times for our Visibility-Aware Param and FlexPara are the same in Table 1 (~2 seconds).
>
> * **Semantic-aware pipeline.** The semantic-aware pipeline does introduce extra inference cost because it performs three stages at test time: (i) 3D semantic partitioning, (ii) per-part parameterization (running the backbone once per part), and (iii) aggregation/UV packing. These additional steps (segmentation, multiple backbone invocations, and packing) explain the larger inference time reported in Table 3 and for the combined pipeline (semantic+visibility-aware) in Table 1. A simple and effective mitigation is to parallelize the per-part backbone calls by using multi-threading during the inference, which would significantly reduce inference time.
>
> ### 2- Training time
>
> As requested by the reviewer, we now added two tables reporting training times: Table 8 (semantic-aware pipeline) and Table 9 (visibility-aware pipeline), and compare both with FlexPara. Below is our analysis of those numbers.
>
> * **Semantic-aware pipeline (Table 8).** Counterintuitively, our semantic-aware pipeline achieves shorter training time than FlexPara across different 3D meshes. This is because the mesh is partitioned into many smaller submeshes: although our semantic-aware pipeline calls the backbone multiple times (once per semantic part), which might suggest longer training, this is compensated by the fact that the 3D segmented parts are much smaller than the original input mesh used by FlexPara. Hence, training on many small submeshes is faster than training a single model on a large, complex mesh with many vertices and faces. This empirical observation is consistent with our careful time measurement of each stage in the semantic-aware pipeline: learning UV parameterization for small 3D parts is much faster than doing the same for large meshes. More importantly, the reduction in per-part training cost outweighs the multiplicative overhead of multiple backbone runs, which results in shorter overall training time for the semantic-aware pipeline in our experiments. This is why Table 8 shows shorter training times for our semantic-aware pipeline compared to FlexPara in our experiments.
>
>
> * **Visibility-aware pipeline (Table 9).** The visibility-aware pipeline trains slower than FlexPara across different meshes. This is expected because, compared to the base neural backbone (i.e., FlexPara), our visibility-aware pipeline introduces per-iteration overheads: (i) per-vertex ambient occlusion (AO) computation, (ii) candidate UV generation via the backbone, (iii) seam/boundary extraction in UV space, and (iv) AO-weighted loss computation. The most expensive stage is seam extraction which accounts for approximately 90% of the additional training time. Specifically, as described in our response to Question 1 of Reviewer Fuqi, for each vertex we inspect its 3–5 neighbors, compare mapped UV positions, and evaluate Equation (5) of the paper to detect UV discontinuities. This neighbor-checking and distance computation scales with vertex count and dominates the additional training cost (~90% of the overhead), making visibility-aware training significantly heavier for complex meshes.

---

> ### Author Response · Authors · 2025-11-21
> **Response to Reviewer Gh6M - Question 1 (part 2)**
>
> ### 3- Practical notes and mitigation
>
> To mitigate the computational burdens discussed above, we propose two practical solutions (these have been now added to Appendix A.3 and highlighted in blue):
>
> * **Parallelize per-part backbone calls (semantic-aware pipeline).** In our current implementation the neural backbone is executed sequentially for each semantic part. Switching to a multi-threaded or multi-GPU implementation to run per-part backbone calls in parallel would substantially reduce both inference and training times.
>
> * **Optimize seam detection (visibility-aware pipeline).** The seam-detection stage is the main training-time bottleneck. Algorithmic and engineering improvements, e.g., spatial hashing, cached neighborhood lookups, and GPU-accelerated neighbor/distance checks, can significantly reduce this overhead. We note these optimizations as straightforward improvements and sensible directions for future work.
>
> We have now (i) added the above mitigation strategies and timing explanations to Appendix A.3, (ii) introduced Tables 8–9 with training-time comparisons, and (iii) highlighted these changes in blue in the revised manuscript. We thank the reviewer for prompting this clarification. We believe these additions improve reproducibility and clarity.
>
> ---
>
> ### Local References:
>
> [1] Srinivasan, Pratul P., et al. "Nuvo: Neural uv mapping for unruly 3d representations." European Conference on Computer Vision. Cham: Springer Nature Switzerland, 2024.
>
> [2] Zhao, Yuming, et al. "FlexPara: Flexible Neural Surface Parameterization." arXiv preprint arXiv:2504.19210 (2025).
>
> [3] Li, Minchen, et al. "Optcuts: Joint optimization of surface cuts and parameterization." ACM transactions on graphics (TOG) 37.6 (2018): 1-13.

---

### Official Review · Reviewer_fuqi · 2025-10-31

**Soundness:** 3
**Presentation:** 3
**Contribution:** 3
**Rating:** 8
**Confidence:** 3

**Summary:**

This paper proposes a fully differentiable and unsupervised framework for 3D mesh parameterization that jointly accounts for semantic consistency and visibility awareness.
The method introduces two key objectives:

(1) a semantic-aware objective, which leverages the Shape Diameter Function (ShDF) to segment the mesh into semantically meaningful regions and then learns UV parameterizations for each region in an unsupervised manner;

(2) a visibility-aware objective, which employs a differentiable Ambient Occlusion (AO) loss to guide seam placement toward less visible areas, thereby improving texture continuity and perceptual quality.

Experimental comparisons with FlexPara, OptCuts, Blender SmartUV, Maya, and xatlas show improvements in semantic alignment, seam invisibility, and geometric distortion.

**Strengths:**

1.The paper is well motivated, aiming to address the challenges of 3D asset generation and to narrow the quality gap between generated assets and artist-created 3D models.

2.The use of off-the-shelf tools to obtain part segmentation information for UV parameterization is both reasonable and effective.

3.Leveraging geometric properties, such as Ambient Occlusion (AO), to determine UV cut seams in a differentiable manner is novel and demonstrates clear effectiveness.

**Weaknesses:**

1.The authors should clarify their specific contributions, as providing coarse semantic information and applying an alpha-cut strategy to optimize per-vertex or per-face labels have already been explored in prior 3D segmentation works. Moreover, the UV parameterization learning approach appears largely adapted from existing methods.

2.Since the semantic partitioning relies on off-the-shelf tools and involves hard segmentation of the mesh into parts before subsequent processing, it is difficult to regard the proposed method as a fully differentiable framework.

3.The proposed UV chart aggregator is straightforward but not very effective. It can lead to significant wasted space on UV maps and may compromise several desirable properties of UV parameterization, such as continuity and uniform distortion.

**Questions:**

The method for determining cut seams appears to be crucial in the visibility-aware UV parameterization, yet Eq. (5) does not clearly explain how the seams are effectively identified. Could the authors elaborate on how this formulation contributes to seam detection and clarify its effectiveness in guiding the visibility-based UV cutting process?

---

> ### Author Response · Authors · 2025-11-21
> **Response to Reviewer fuqi - Weakness 1**
>
> We thank the reviewer for this helpful comment. We agree that alpha-cut strategies for converting soft per-vertex or per-face labels into hard partitions have been explored in prior 3D segmentation work (e.g., [1]). We also emphasize that the UV neural backbone architecture we adopt is not claimed as a novelty of this paper. Instead, our primary contribution is the design and integration of semantic-aware and visibility-aware objectives as downstream tasks built on top of this established geometric neural foundation.
>
> Specifically, our contributions consist of three main components. First, for semantic awareness we propose a novel “partition-and-parameterize” strategy: we first partition the input mesh into smaller, semantically meaningful 3D parts, then learn geometry-preserving UV parameterizations for each part independently, and finally aggregate those per-part parameterizations into a unified UV atlas. To our knowledge, this specific pipeline, partitioning into semantic parts and then learning per-part parameterizations followed by aggregation, has not been proposed before. Second, we introduce a visibility-aware objective based on ambient occlusion: this objective extracts seam candidates and guides seams toward less-visible (more occluded) surface regions to reduce seam artifacts in rendered textures. Finally, we combine these two ideas into a single framework that jointly optimizes semantic-awareness and visibility-awareness alongside traditional geometry-preserving losses. While the backbone itself follows prior geometry-preserving work, the downstream objectives and their integration constitute the novel and practical contributions of this paper.
>
> Also, as reviewer vZwh pointed out similar comment, to make our primary contribution explicit, we now have added the following sentence at the start of Section 3.2 (highlighted in blue the revision):
>
> *"We emphasize that this stage adopts an existing, previously proposed low-distortion parameterization backbone; we do not claim novelty for the backbone itself. Instead, our primary contribution is the design and integration of semantic-aware and visibility-aware objectives as downstream tasks built on top of this established geometric foundation."*
>
> We also expanded the surrounding text to better emphasize that the novelty of the paper lies in these downstream objectives and their interaction with geometry preservation. Thank you again for the constructive suggestion.
>
> ---
> ### Local References:
>
> [1] Shapira, Lior, Ariel Shamir, and Daniel Cohen-Or. "Consistent mesh partitioning and skeletonisation using the shape diameter function." The Visual Computer 24.4 (2008): 249-259.

---

> ### Author Response · Authors · 2025-11-21
> **Response to Reviewer fuqi - Weakness 2**
>
> We thank the reviewer for this careful observation. Your understanding is correct. In our current implementation of the semantic-aware UV-parameterization pipeline, the segmentation stage involves hard decisions that partition the mesh into discrete parts. As a result, the proposed pipeline is not fully end-to-end differentiable. If a smooth, learning-based segmentation method (i.e., one that produces soft, differentiable partitions) is used in place of the current hard partitioning, the pipeline would become fully differentiable. We have not implemented such an alternative yet, but this is an interesting and promising direction for future work.
>
> We appreciate the reviewer’s careful reading.

---

> ### Author Response · Authors · 2025-11-21
> **Response to Reviewer fuqi - Weakness 3**
>
> We thank the reviewer for this constructive feedback. This observation is indeed correct. Reviewer vZwh also raised a similar comment. We already addressed it briefly at the end of Section 3.3. Our current aggregation strategy is intentionally simple, meant to demonstrate a clear and interpretable way of combining UV charts after parameterizing the partitioned 3D parts in our semantic-aware pipeline. We agree that more sophisticated aggregation schemes could utilize UV space more efficiently.
>
> Therefore, to further improve our semantic-aware UV parameterization pipeline's output, a simple enhancement is to post-process the UV charts by jointly optimizing two key UV metrics: texel density and UV space utilization. Higher texel density improves the quality of the projected 3D texture, and maintaining consistent texel density across charts ensures uniform visual fidelity. At the same time, efficient space utilization maximizes the use of available texture memory.
>
> Hence, to analyze the effect of more advanced UV-chart packing (aggregator) strategies (different from the one proposed in Sec.3.3), we perform an ablation study and evaluate several UV-packing techniques from three widely used commercial tools including, Autodesk Maya, Houdini, and Blender. We use a 3D cow mesh shown in Fig.7 to evaluate texel density and UV-space utilization of UV parameterizations packed using different UV packing strategies from the three commercial tools. Specifically, we apply our semantic-aware UV parameterization framework as follows: (i) the 3D cow is first partitioned into seven semantic 3D parts, (ii) then the geometry-preserving neural backbone is applied separately to each part to learn UV parameterizations for each 3D part, (iii) instead of using our default UV packing (presented in Sec.3.3), we apply different UV packing strategies from the three commercial tools, and (iv) lastly, for each packing strategy, we compute two UV metrics, namely texel density and UV-space utilization, for the UV parameterizations resulting from the three different UV packing methods. We have now added Table 5 to the revised manuscript. This table reports the resulting values for both metrics to illustrate how each tool performs on a 3D cow mesh example. Also, we have now added Fig. 7 to the revised manuscript as a qualitative comparison of this ablation study. Fig. 7 qualitatively compares the three commercial tools, applying their automatic packing and layout features to the same unwrapped mesh of 3D cow.
>
>
> A detailed description of these strategies is provided in Appendix A.3 of the revised manuscript, and the new material is highlighted in blue.
>
>
> Finally, we emphasize the modularity of our pipeline: In the semantic-aware UV parameterization pipeline, the UV aggregator (packing) stage is interchangeable. This means that if a user prefers a different UV packing method (e.g., Autodesk Maya, Houdini, Blender, etc.), it can be plugged into our semantic-aware UV pipeline with minimal modifications.
>
>
> We appreciate the reviewer’s constructive and valid feedback. It motivated us to try multiple UV-packing strategies and compare them both qualitatively and quantitatively. These experiments demonstrate the generalizability of our semantic-aware UV-parameterization framework (it works with different packing strategies) and, importantly, provide users and practitioners with verified insights about how different packing choices affect the final UV parameterization.

---

> ### Author Response · Authors · 2025-11-21
> **Response to Reviewer fuqi - Question 1**
>
> We thank the reviewer for this careful comment. Although Section 3.4 already describes the steps for extracting cutting seams (under “Cutting Seam Extraction in UV Space”), we agree that the role of Equation (5) was not explained with sufficient clarity. To address this, we have added a detailed intuition and explanation around Equation (5) in the revised manuscript (highlighted in blue). Below we summarize the key idea:
>
> During training, at each iteration we must identify the cutting-seam (boundary) points in UV space in order to (i) evaluate their ambient occlusion values and (ii) compute the visibility-aware loss in Equation (7). Minimizing this loss encourages the model to learn UV parameterizations whose seams lie in less-visible (low-AO) regions.
>
> Equation (5) provides a mechanism for detecting these seam points. Rather than locating seams directly on the 3D mesh, we detect them in the UV domain based on how local neighborhoods deform during unwrapping. Specifically, for each 3D vertex we know both its 3D coordinates and its corresponding UV coordinates at the current iteration. Therefore, we examine the 3D neighbors of each vertex and compare their mapped UV positions. If a pair of vertices are close neighbors on the 3D surface but become separated by more than a threshold distance in UV space, this indicates that the surface has been cut between them. We therefore mark such UV points as seam (boundary) points.
>
> This criterion is grounded in the fact that any valid unwrapping must introduce discontinuities (i.e., seams): vertices that are adjacent on the 3D surface become far apart in the UV plane after the mesh is cut open. Equation (5) formalizes this idea by detecting precisely those locations where this large separation occurs.
>
> We hope this clarifies the role and intuition of Equation (5). We have added this explanation to the manuscript near Equation (5). We thank the reviewer for raising this point and helping us improve the clarity of the exposition.

---

### Official Review · Reviewer_vZwh · 2025-11-01

**Soundness:** 3
**Presentation:** 3
**Contribution:** 3
**Rating:** 6
**Confidence:** 3

**Summary:**

This paper presents an unsupervised, differentiable framework designed to automate 3D mesh parameterization (UV mapping), addressing this critical and often manual bottleneck in 3D content creation. The core contribution is augmenting standard geometry-preserving UV learning with two novel perceptual criteria: semantic awareness and visibility awareness.

The semantic-aware pipeline utilizes a partition-and-parameterize strategy, first segmenting the mesh into meaningful 3D parts using a Shape Diameter Function (ShDF), and then independently parameterizing and aggregating these parts into a unified atlas.

Conversely, the visibility-aware objective leverages ambient occlusion (AO) as a differentiable proxy for viewer exposure, actively steering the placement of cutting seams toward regions of low visibility. By minimizing this AO-weighted seam loss, the framework ensures that UV cutting seams are less likely to be observed, thereby reducing perceptible artifacts after texturing and rendering.

Overall, qualitative and quantitative comparisons confirm that the proposed method generates UV atlases that maintain low geometric distortion while achieving better semantic coherence and superior, perceptually favorable seam placement compared to existing baselines.

**Strengths:**

- The paper is easy to follow. The framework successfully transitions from the standard foundation of geometry-preserving mesh parameterization learning to its specific novel contributions: semantic awareness and visibility awareness.

- The research includes comprehensive comparisons against state-of-the-art baselines, encompassing both recent open-source learning methods (e.g., FlexPara, xatlas) and established industrial software tools (e.g., Autodesk Maya and Blender’s SmartUV)

- The proposed semantic-aware and visibility-aware parameterizations are effective.

  - The semantic-aware pipeline generates practical, high-utility UV atlases by employing a partition-and-parameterize strategy based on the ShDF. This produces UV charts that align with meaningful 3D parts, making them cleaner, more interpretable, and directly beneficial for downstream tasks such as texture painting, editing, and transfer.
  - The visibility-aware objective significantly improves perceptual quality by actively minimizing an (AO–weighted loss, successfully steering cutting seams toward less-exposed, more occluded regions. Using AO as a proxy supervision signal is a concise and elegant way to control visibility.

**Weaknesses:**

- The methodology’s initial phase, "geometry-preservation mesh-parameterization learning" (Sec. 3.2), adopts a network architecture based on the bi-directional cycle mapping backbone proposed in prior work (e.g., Zhao et al., 2025). This phrasing may lead readers to believe Stage I is a novel geometry parameterization method, rather than emphasizing that the true contribution lies in integrating semantic-aware and visibility-aware objectives as downstream tasks atop an existing geometric backbone. The manuscript could improve clarity by explicitly framing Stage I as an adoption of a known geometry-preserving backbone (e.g., FlexPara) and focusing the novelty on the downstream integration of the semantic and visibility objectives.
- The ablation comparing the Shape Diameter Function (ShDF) partitioner to the SAMesh baseline is confusing. Quantitatively, SAMesh output is treated as GT for semantic metrics (Hamming Distance and Rand Index). Yet, the ablations show that the proposed ShDF partitioner yields UV parameterizations that are geometrically superior (more conformal and more equiareal) than those using SAMesh. Since ShDF is primarily geometry-driven (local thickness), the paper lacks external validation against industry-standard or human-annotated semantic datasets (e.g., PartNet, PartNeXt) to confirm whether ShDF’s partitions align with true human semantic decomposition.

**Questions:**

- The final stage of the semantic pipeline employs an "intentionally simple and deterministic" aggregation and packing strategy, placing UV islands into a $G\times G$ regular grid. While this is simple and beneficial for texture editing and transfer workflows, the authors note it can be replaced by "more advanced packing solvers." This choice implies the final UV atlas likely suffers from suboptimal UV space utilization or packing efficiency—a critical metric in 3D production that is not reported in the quantitative analysis.

- Although the authors claim that adding perceptual objectives causes only "minor changes" in geometry-preserving metrics, the quantitative results show a noticeable drop in equiareality (area preservation) when the visibility objective is enabled—from 0.6759 (Base Neural UV Arch., i.e., FlexPara) to 0.6093 (Ours—Visibility-Aware Param). While the authors consider this “minor,” such degradation may be significant for applications requiring precise triangle area preservation (e.g., lightmap baking). The paper does not provide an in-depth analysis of whether this loss of geometric fidelity is the minimal necessary cost to achieve perceptually superior seam placement.

Could the authors provide more in-depth analysis and discussion on the above two points?

---

> ### Author Response · Authors · 2025-11-21
> **Response to Reviewer vZwh - Weakness 1**
>
> We thank the reviewer for this helpful reading. Your understanding is exactly correct. To address your suggestion and make our primary contribution explicit, we have added the following sentence at the begining of Section 3.2 (highlighted in blue in the revision):
>
> *"We emphasize that this stage adopts an existing, previously proposed low-distortion parameterization backbone; we do not claim novelty for the backbone itself. Instead, our primary contribution is the design and integration of semantic-aware and visibility-aware objectives as downstream tasks built on top of this established geometric foundation."*
>
> We have also expanded the surrounding text to further clarify that the novelty of the paper lies in these downstream objectives and their interaction with geometry preservation. Thank you again for the constructive suggestion.

---

> ### Author Response · Authors · 2025-11-21
> **Response to Reviewer vZwh - Weakness 2**
>
> We thank the reviewer for raising this important point. To clarify and remove any confusion, we explain two distinct comparisons made in the manuscript:
>
> **1- Geometry-preservation comparison (ShDF vs. SAMesh).**
>
> In the appendix, we compare ShDF and SAMesh strictly on geometry-preservation metrics (conformality and equiareality): qualitatively in Figures 14–15 and quantitatively in Table 5. These results show that ShDF partitions objects into semantically meaningful parts while better preserving geometric properties (lower conformal and area distortion) than SAMesh. This finding is important: although our overall pipeline adds semantic- and visibility-aware objectives (which necessarily trade off some geometry preservation), ShDF provides a stronger starting point for keeping distortion low within that pipeline, i.e., it yields less geometric degradation when used in our method compared to SAMesh.
>
> Also, comparing ShDF with SAMesh or any other 3D segmentation method from the perspective of semantic-awareness is not the focus of this work. Such comparisons have already been addressed in the SAMesh paper and the broader 3D segmentation literature. We propose ShDF as the 3D partitioning stage in our pipeline because it is simpler and more straightforward than many alternatives and, as noted above, it achieves better geometry-preserving parameterizations when used in our semantic-aware pipeline. While ShDF can be replaced by any other 3D segmentation method, a detailed comparison of 3D partitioning methods is outside the scope of this paper.
>
> Finally, following a suggestion from reviewer dJSn, we have also incorporated PartField (a modern learning-based method) into our ablation study as an additional 3D segmentation baseline. Figures 14–15 and Table 5 in the revised manuscript now compare ShDF, SAMesh, and PartField. Including PartField strengthens the evaluation of our partitioning strategy and further shows that our pipeline is compatible with state-of-the-art segmentation methods. These new results help clarify the trade-offs between semantic coherence and geometry preservation across different partitioners.
>
> **2- Semantic-awareness comparison (our pipeline vs. baselines).**
>
> Table 3 in the main paper compares the full semantic-aware UV pipeline (which uses ShDF in the 3D segmentation stage; see Sec. 3.3 and Appendix A.1) against baseline UV methods under semantic metrics (Hamming distance and Rand Index). This is not a ShDF-vs-SAMesh comparison; rather, it evaluates whether our pipeline produces UV atlases with higher semantic consistency than the baselines. For these semantic comparisons we used SAMesh as the reference segmentation (ground truth) because, at the time of our experiments, SAMesh was the state-of-the-art automatic learning-based 3D segmentation method. We agree that other ground-truth sources (e.g., PartNet, PartNetXt) are valid alternatives and could also be used to compute the semantic metrics; using different segmentation references is straightforward and we will explore this in future work.
>
> However, to address the reviewer concern about the semantic meaningfulness of ShDF’s partitions and to show that our UV parameterization is valid and human-interpretable, we ran a user study asking participants several questions about the semantic meaningfulness of UV parameterizations produced by our method and four baselines (Autodesk Maya, Blender, xatlas, and FlexPara). We have now added the user-study results (with 65 participants) to the paper in Table 2 and Figure 16. The results show a strong user preference for the semantic meaningfulness of the UV parameterizations produced by our method over the compared baselines.
>
> Finally, we emphasize the modularity of our pipeline: the 3D segmentation stage is interchangeable. This means that if a user prefers a different segmentation method (e.g., SAMesh or a PartNet-derived segmentation), it can be plugged into our semantic-aware UV pipeline without modification. Thank you for the suggestion, it helped us make the comparisons and their interpretation much clearer.

---

> ### Author Response · Authors · 2025-11-21
> **Response to Reviewer vZwh - Question 1**
>
> We thank the reviewer for this constructive feedback. This observation is indeed correct, and we already addressed it briefly at the end of Section 3.3. Our current aggregation strategy is intentionally simple, meant to demonstrate a clear and interpretable way of combining UV charts after parameterizing the partitioned 3D parts in our semantic-aware pipeline. We agree that more sophisticated aggregation schemes could utilize UV space more efficiently.
>
> Therefore, to further improve our semantic-aware UV parameterization pipeline's output, a simple enhancement is to post-process the UV charts by jointly optimizing two key UV metrics: texel density and UV space utilization. Higher texel density improves the quality of the projected 3D texture, and maintaining consistent texel density across charts ensures uniform visual fidelity. At the same time, efficient space utilization maximizes the use of available texture memory.
>
> Hence, to analyze the effect of more advanced UV-chart packing (aggregator) strategies (different from the one proposed in Sec.3.3), we perform an ablation study and evaluate several UV-packing techniques from three widely used commercial tools including, Autodesk Maya, Houdini, and Blender. We use a 3D cow mesh shown in Fig.7 to evaluate texel density and UV-space utilization of UV parameterizations packed using different UV packing strategies from the three commercial tools. Specifically, we apply our semantic-aware UV parameterization framework as follows: (i) the 3D cow is first partitioned into seven semantic 3D parts, (ii) then the geometry-preserving neural backbone is applied separately to each part to learn UV parameterizations for each 3D part, (iii) instead of using our default UV packing (presented in Sec.3.3), we apply different UV packing strategies from the three commercial tools, and (iv) lastly, for each packing strategy, we compute two UV metrics, namely texel density and UV-space utilization, for the UV parameterizations resulting from the three different UV packing methods. We have now added Table 5 to the revised manuscript. This table reports the resulting values for both metrics to illustrate how each tool performs on a 3D cow mesh example. Also, we have now added Fig. 7 to the revised manuscript as a qualitative comparison of this ablation study. Fig. 7 qualitatively compares the three commercial tools, applying their automatic packing and layout features to the same unwrapped mesh of 3D cow.
>
>
> A detailed description of these strategies is provided in Appendix A.3 of the revised manuscript, and the new material is highlighted in blue.
>
>
> Finally, we emphasize the modularity of our pipeline: In the semantic-aware UV parameterization pipeline, the UV aggregator (packing) stage is interchangeable. This means that if a user prefers a different UV packing method (e.g., Autodesk Maya, Houdini, Blender, etc.), it can be plugged into our semantic-aware UV pipeline with minimal modifications.
>
>
> We appreciate the reviewer’s constructive and valid feedback. It motivated us to try multiple UV-packing strategies and compare them both qualitatively and quantitatively. These experiments demonstrate the generalizability of our semantic-aware UV-parameterization framework (it works with different packing strategies) and, importantly, provide users and practitioners with verified insights about how different packing choices affect the final UV parameterization.

---

> ### Author Response · Authors · 2025-11-21
> **Response to Reviewer vZwh - Question 2**
>
> We thank the reviewer for this careful comment about Table 7 in the revised paper. We fully agree that there are unavoidable trade-offs when optimizing multiple geometry-preservation objectives together, and that these trade-offs become more pronounced once higher-level goals (semantic- and visibility-awareness) are added. This is a well-known tension in the parameterization literature: even when optimizing only geometric objectives (conformality, equiareality, stretch, bijectivity, etc.), it is generally impractical to obtain the best possible value for every metric simultaneously. Adding complex, high-level objectives makes the multi-objective optimization even harder, so some loss in geometry-preservation is expected and was anticipated at the beginning of this work.
>
> To quantify these effects, we included an ablation in the Appendix (Table 7) that compares our proposed pipelines to the baseline (labelled “Base Neural UV Arch” in Table 7, i.e., FlexPara which is state-of-the-art for geometry preservation). We have now expanded the Table 7 caption and Appendix A.3 (highlighted in blue in the revision) with the following analysis:
>
> **1- Clarification of method names.**
>
> In Table 7: “Semantic-Aware Param” refers to the pipeline from Sec. 3.3 (visualized in Fig. 2); “Visibility-Aware Param” refers to the pipeline from Sec. 3.4 (Alg. 1); and “Visibility+Semantic-Aware Param” denotes the combined pipeline that applies both sets of objectives.
>
> **2- Semantic-Aware Param / Visibility+Semantic-Aware Param vs. baseline.**
>
> When comparing conformality and equiareality, the Semantic-Aware and the combined Visibility+Semantic-Aware pipelines lose little equiareality relative to the baseline and even show slight improvement in conformality in some cases. Intuitively, this is because the semantic-aware stage first partitions the mesh into multiple semantically meaningful subparts. Learning a geometry-preserving parameterization on smaller, simpler 3D patches is easier: the network can better satisfy geometry-preservation while also enforcing semantic consistency. Therefore, semantic partitioning provides a natural way to reduce distortion without heavily sacrificing geometry metrics.
>
> **3- Visibility-Aware Param vs. baseline.**
>
> However, the visibility-aware pipeline does not partition the mesh. Instead, it steers seams to less-visible (more occluded) regions while producing one global atlas (a single large chart). Unwrapping and optimizing a single global chart to satisfy both geometry-preservation and visibility objectives is significantly harder. Consequently, the Visibility-Aware pipeline shows a larger drop in equiareality compared to the baseline and compared to the semantic-aware pipelines.
>
> **4- Combined pipeline as a sweet spot.**
>
> The combined Visibility+Semantic-Aware pipeline aims to obtain the best of both worlds: (i) partition the mesh into smaller parts to retain geometry preservation, and (ii) steer seams toward less-visible regions within those parts. This combined approach yields a favorable trade-off: it loses far less equiareality (0.6369) than the pure visibility-aware pipeline (0.6093) while still improving seam visibility and semantic consistency.
>
> We have now added this analysis to the caption of Table 7 and expanded Appendix A.3 to include the additional discussion (all highlighted in blue). Thank you, this suggestion helped us make the trade-offs and their interpretation much clearer.

---

### Official Review · Reviewer_dJSn · 2025-11-01

**Soundness:** 3
**Presentation:** 2
**Contribution:** 3
**Rating:** 6
**Confidence:** 4

**Summary:**

This work proposes a method for predicting a UV parameterization for a mesh that not only respects common geometric properties such as minimizing distortion and cuts, but also tries to keep semantically related components nearby in UV space and place cuts / seams in low visibility regions. To accomplish this, the method uses the ShDF to segment the mesh into semantic components and then uses ambient occlusion as a proxy for visibility. Each semantic segment is independently parameterized and then all segments are recombined and packed into a single atlas. The method is evaluated against several baselines both quantitatively with comparison figures and quantitatively using specific semantic and visibility metrics as well as standard distortion metrics. Ablations of several key method components are also included.

**Strengths:**

- This work identifies a gap in the current state of parameterization methods. It then addresses this by proposing a method producing parameterizations that group semantic components and place seams at low visibility regions.
- The soft seam membership introduced to make the cutting differentiable is an interesting contribution and can be used in future research aiming to take a learning based approach to UV parameterization.

**Weaknesses:**

- Distortion looks higher on this method in Fig 3 and is quantitatively worse in Tab. 1 and Tab. 2., indicating that this approach sacrifices distortion for semantics and less visible seams
- As noted in this paper, a good parameterization is subjective and varies a lot artist to artist. Given this, it would be beneficial to include a user study where actual 3D artists can evaluate the quality of the parameterizations.
- Quantitative metrics for seam visibility evaluate using the same ambient occlusion that this method optimizes for. While AO might be a good metric for visibility, it is likely not perfect in all situations and thus optimizing for it and then measuring on it feels unfair.

**Questions:**

- Based on my interpretation of eq 7 it seems the relationship is backward and the $AO_i$ should be $(1 - AO_i)$ in the numerator?
- Have authors explored using other modern semantic segmentation methods? I.e Partfield [1]? I see authors have ablated the use of SAMesh but perhaps Partfield would work better. I wonder if on harder shapes learning based segmentation methods would work better than ShDF. Could the authors provide more intuition for why the ShDF segmentations produced more conformal and equiareal parameterizations?

References: [1] Liu, Minghua, et al. "Partfield: Learning 3d feature fields for part segmentation and beyond." Proceedings of the IEEE/CVF International Conference on Computer Vision. 2025.

---

> ### Author Response · Authors · 2025-11-21
> **Response to Reviewer dJSn - Weakness 1**
>
> We thank the reviewer for this important observation. We agree that our method produces an increase in the standard distortion metrics (as shown in Fig. 3 and Tables 1 and 3). However, this is a trade-off that is expected when optimizing multiple objectives during the learning process. Specifically, our optimization introduces explicit semantic-alignment and visibility-awareness terms that encourage semantically consistent charts and reduce conspicuous seams. In contrast, existing geometry-preserving methods (e.g., Autodesk Maya, Blender, xatlas, and FlexPara) focus solely on traditional distortion and metric objectives. Therefore, they score better on those metrics but do not provide the semantic- or visibility-related guarantees offered by our method.
>
> We note that these semantic and visibility properties are often more valuable in downstream tasks such as mass production, semantic editing, and 3D asset reuse, where consistent UV semantics and low seam visibility significantly reduce manual effort. Importantly, the trade-off is controllable in our framework: the relative emphasis between distortion and semantic/visibility objectives can be adjusted through tunable weights and hyperparameters associated with the different loss functions in Equations (1) and (8), so users can shift toward lower distortion when preferred.
>
> We have now added this analysis to Section 4.2 under the “Geometry-Preservation Evaluation” to include this additional discussion (all highlighted in blue).
>
> We hope these explanations clarify when and why a small increase in distortion is an acceptable, and often desirable, compromise.

---

> ### Author Response · Authors · 2025-11-21
> **Response to Reviewer dJSn - Weakness 2**
>
> We thank the reviewer for this helpful suggestion. We also agree that evaluating the parameterizations by 3D artists can give us a valid metric and real insight about how high-quality our parameterizations are and how far we still are from industry-level parameterizations.
>
> To this end, and to evaluate semantic- and visibility-awareness of the proposed method, we conducted a user-preference study with 65 participants. Each participant completed 11 comparisons between textured 3D shapes and UV parameterizations produced by our method, FlexPara [1], OptCuts [2], Autodesk Maya [3], Blender [4], and xatlas [5]. For each comparison, participants rated each result according to three visual criteria: (i) texture-pattern smoothness/continuity, (ii) semantic alignment (how well colors correspond to meaningful parts), and (iii) seam visibility (how well seams are placed in occluded/less-obvious regions). The semantic-aware and visibility-aware evaluations comprised 5 and 6 questions, respectively, for a total of 11 comparisons per participant. We have now added Table 2 to the revised paper reporting the percentage of participant preferences for each method. As the table shows, our proposed method is strongly preferred by users over the baselines. Examples of questions in our user-study questionnaire are shown in Fig. 16 in the revised paper.
>
> Please note that we distributed the questionnaire to two groups: 3D artists and general participants. Recruiting professional 3D artists and obtaining their feedback has taken longer than anticipated and could not be completed within a week. At this stage, we have collected responses from 65 general participants (including graduate students with computer science and engineering backgrounds). For this group, the questionnaire was designed to be accessible, with questions that do not require technical expertise. We expect to receive the artists’ responses in the coming days, and we anticipate having them available before the conference discussion period ends. These additional results will be incorporated into the paper.
>
> We thank the reviewer again for this constructive feedback. It helped us evaluate our work with an eye toward industry-level practice.
>
> ---
>
> ### Local References:
>
> [1] Zhao, Yuming, et al. "FlexPara: Flexible Neural Surface Parameterization." arXiv preprint arXiv:2504.19210 (2025).
>
> [2] Li, Minchen, et al. "Optcuts: Joint optimization of surface cuts and parameterization." ACM transactions on graphics (TOG) 37.6 (2018): 1-13.
>
> [3] Autodesk, Inc. Autodesk Maya (Version 2025.x). Autodesk, Inc., 2025. URL https://www.autodesk.com/products/maya/overview. Accessed: 2025-09-11
>
> [4] Blender Foundation. Blender – a 3D creation suite (Version 4.5). Blender Foundation, 2025. URL https://www.blender.org. Accessed: 2025-09-11.
>
> [5] jpcy. xatlas: Mesh parameterization / uv unwrapping library. https://github.com/jpcy/xatlas, 2025. Accessed: 2025-09-11.

---

> ### Author Response · Authors · 2025-11-21
> **Response to Reviewer dJSn - Weakness 3**
>
> We thank the reviewer for this careful comment. We fully agree that training the model on ambient occlusion (AO) and then evaluating it again using AO is unfair. Unfortunately, most visibility metrics we are aware of are closely related to the ray-tracing idea behind AO, so finding a completely independent, well-established metric is not straightforward. However, one promising alternative we considered (and discussed among the authors) is rendering each object from a set of visible camera viewpoints and computing a metric on the rendered images to measure how well seams are hidden from those viewpoints. This approach would provide viewpoint-specific, image-space validation of seam visibility. We coulde explore a related idea to compute per-view heat maps to capture seam visibility. But those maps still relied on AO-like computations.
>
> To address the reviewer’s excellent point and provide an external, human-centered validation of seam visibility, we therefore ran a user-preference study. We believe that asking human observers to judge which seams are less visible provides a fair and complementary evaluation of visibility-awareness compared to AO-based measures, and we report those user-study results in the revised manuscript.
>
> To this end, we conducted a user-preference study with 65 participants, as also we explained this in the answer to the reivewer's "Weakness 2" comment. In particular, for visibility-awareness, each participant completed 6 comparisons between textured 3D shapes and UV parameterizations produced by our method, FlexPara [1], and OptCuts [2]. For each comparison, participants rated each result according to two visibility-aware criteria: (i) texture-pattern smoothness/continuity, and (ii) seam visibility (how well seams are placed in occluded/less-obvious regions). For each of six questions, we rendered the 3D mesh from different visible camera viewpoints with the UV parameterization and the cutting seams visualized on the mesh surface. Then, the participants evaluated each result according to the two criteria described above. We have added Table 2 to the revised paper reporting the percentage of participant preferences for each method. As the table shows, our visibility-aware method (based on AO) is strongly preferred by users. 92% of responses preferred our method in comparisons against FlexPara and OptCuts. This result indicates that the seams produced by our method more often lie in occluded regions, which consequently yields smoother and more continuous textured surfaces compared to FlexPara and OptCuts. Examples of questions in our user-study questionnaire are shown in Fig. 16 in the revised paper.
>
> We hope the clarification above resolves the reviewer’s concern.
>
>
> ---
> ### Local References:
>
> [1] Zhao, Yuming, et al. "FlexPara: Flexible Neural Surface Parameterization." arXiv preprint arXiv:2504.19210 (2025).
>
> [2] Li, Minchen, et al. "Optcuts: Joint optimization of surface cuts and parameterization." ACM transactions on graphics (TOG) 37.6 (2018): 1-13.

---

> ### Author Response · Authors · 2025-11-21
> **Response to Reviewer dJSn - Question 1**
>
> We thank the reviewer for catching this inconsistency. Equation (7) itself is correct. By our convention in Equation (4), AO(p)=1 denotes a fully visible point and  AO(p)=0 denotes a fully occluded point. Therefore, minimizing Equation (7) naturally drives seams toward regions with lower AO values (i.e., more occluded areas), which is exactly the intended behavior.
>
> However, the confusion arises from a typo in the sentence immediately before Equation (7), which incorrectly states “where 0 indicates fully visible and 1 indicates fully occluded.” This should instead read: “where 0 indicates fully occluded and 1 indicates fully visible.”
>
> We have corrected this typo in the revised manuscript (highlighted in blue) and added an explicit cross-reference to Equation (4) to avoid further ambiguity. We appreciate the reviewer’s careful reading and attention to detail.

---

> ### Author Response · Authors · 2025-11-21
> **Response to Reviewer dJSn - Question 2**
>
> We thank the reviewer for this comment and for suggesting an additional semantic segmentation method. As requested, we have now added PartField [1] as an alternative 3D partitioning method in our semantic-aware UV pipeline and included it in our ablation study in Appendix A.3. Figures 14-15 and Table 6 in the revised manuscript have been updated to show the qualitative and quantitative results, respectively, for ShDF vs. SAMesh vs. PartField.
>
> To address the reviewer’s two follow-up points:
>
> **(i) On using modern learning-based segmentation.**
>
> State-of-the-art learning-based segmentation methods can produce more semantically coherent partitions than ShDF (as shown by works such as SAMesh). Therefore, any improvements in that literature can directly enhance the semantic-awareness of our pipeline, since segmentation is the first stage of our semantic-aware parameterization (Fig. 2). In other words, a better 3D partitioner is an immediate plug-in improvement for our method.
>
> **(ii) On ShDF yielding more conformal/equiareal parameterizations.**
>
> There is no obvious or general relationship between the choice of segmentation algorithm and the geometry-preservation metrics (conformality and equiareality) of the final UV parameterization. This is because of the fact that existing 3D segmentation methods were developed to produce semantically meaningful parts, not to optimize downstream UV parameterization. Consequently, they do not incorporate UV-specific information. Therefore, from the numbers in Table 6 in the revised paper (ShDF, SAMesh, PartField) we cannot draw a unified, causal conclusion connecting segmentation method type to final UV geometry properties (e.g. conformality or equiareality). That said, we include the segmentation ablation and Table 6 intentionally to inform practitioners that different partitioning methods, while varying in semantic quality, can also influence geometry preservation in the resulting UVs. Reporting these results highlights the important trade-off between semantic-awareness and geometric fidelity. Therefore, practitioners should try different segmentation approaches and choose the one that best fits their application needs.
>
> We appreciate the reviewer’s suggestion: studying the relationship between 3D segmentation choice and downstream UV geometry is an interesting research question in its own right, and we leave it as future work.
>
> ---
> ### Local References:
>
> [1] Liu, Minghua, et al. "Partfield: Learning 3d feature fields for part segmentation and beyond." Proceedings of the IEEE/CVF International Conference on Computer Vision. 2025.

---

### Author Response · Authors · 2025-11-27
**Summary of Revisions to the Manuscript and Responses to Reviewers’ Comments**

We thank the reviewers for their constructive and positive feedback. We have updated the manuscript with the requested additional discussion, details, and experiments, all highlighted in blue in the revised manuscript. We now outline the changes below:

* **[Reviewers dJSn, vZwh, fuqi].** We have clarified all requested details and explanations, including Equations (5) and (7), and the contribution of the work in the main paper (pages 4 and 7).

* **[Reviewers dJSn, vZwh].** To address concerns about how artists’/users’ preferences behave and as an additional external evaluation for both the proposed semantic-aware and visibility-aware UV parameterization pipelines, we conducted a user-preference study with 98 participants (so far). The study details and results are reported at the beginning of Section 4.2, in Table 2 of the main paper, in Table 10 and Figure 16 of the appendix. We distributed the questionnaire to two groups: experts and general participants. Of the 98 participants, 65 are general participants (including graduate students with computer science and engineering backgrounds) and 33 (so far) are experts: 22 software engineers, 3 project managers, 2 product owners, 4 UV-layout artists, and 2 modeling artists, all working in the Media & Entertainment industry for film and games. We are continuing to collect expert responses and expect to add the additional results to the manuscript before the conference discussion period ends.

* **[Reviewers vZwh, fuqi, Gh6M].** To address the reviewers' request to analyze the effect of more advanced UV-chart packing (aggregator) strategies (different from the one proposed in Section 3.3), we performed an ablation study and quantitatively and qualitatively evaluated several UV-packing techniques from three widely used commercial/professional tools: Autodesk Maya [1], Houdini [2], and Blender [3]. Details, discussion, and results are now in Appendix A.3, Table 5, and Figure 7.

* **[Reviewers dJSn, vZwh].** Regarding the effect of different semantic partitioning methods/strategies, and as requested by reviewer dJSn, we added PartField [4] as a modern, state-of-the-art partitioning method to our semantic-aware UV parameterization pipeline. We now qualitatively and quantitatively compare three strong partitioning strategies, PartField [4], ShDF [5], and SAMesh [6], and include a detailed discussion plus qualitative and quantitative results in Appendix A.3, Table 6, Figure 14, and Figure 15.

* **[Reviewers dJSn, vZwh].** To discuss the trade-off when optimizing multiple objectives (geometry-preserving objectives, e.g., conformality and equiareality, versus high-level texture-related objectives such as semantic-awareness and visibility-awareness), we added a detailed analysis and ablation study. The discussion and results appear in Section 4.2, Appendix A.3, and Table 7.

* **[Reviewer Gh6M].** As requested, we clarified the analysis of inference computation time between our method and the baselines (see Table 1 and Table 3). To address the reviewer’s question about training cost, we also added per-mesh training computation times for our pipeline and quantitatively compared them with the state-of-the-art method FlexPara [7]. The discussion and results for training computation time are in Appendix A.3, Table 8, and Table 9.

We appreciate the reviewers’ constructive and insightful feedback, which has helped us improve the quality of the manuscript.

---
### Local References:

[1] Autodesk, Inc. Autodesk Maya (Version 2025.x). Autodesk, Inc., 2025. URL https://www.autodesk.com/products/maya/overview. Accessed: 2025-09-11

[2] Toronto-based SideFX. Houdini – a 3D animation software (Version 21). Toronto-based SideFX, 2025. URL https://www.sidefx.com/products/houdini/. Accessed: 2025-11-20.

[3] Blender Foundation. Blender – a 3D creation suite (Version 4.5). Blender Foundation, 2025. URL https://www.blender.org. Accessed: 2025-09-11.

[4] Minghua Liu, Mikaela Angelina Uy, Donglai Xiang, Hao Su, Sanja Fidler, Nicholas Sharp, and Jun Gao. Partfield: Learning 3d feature fields for part segmentation and beyond. In Proceedings of the IEEE/CVF International Conference on Computer Vision, pp. 9704–9715, 2025.

[5] Shapira, Lior, Ariel Shamir, and Daniel Cohen-Or. "Consistent mesh partitioning and skeletonisation using the shape diameter function." The Visual Computer 24.4 (2008): 249-259.

[6] George Tang, William Zhao, Logan Ford, David Benhaim, and Paul Zhang. Segment any mesh. arXiv preprint arXiv:2408.13679, 2024.

---

### Author Response · Authors · 2025-12-01
**Summary of the Completed User-Preference Study**

As mentioned in our previous response, we were working hard to complete the user-preference study. We are now writing to let all reviewers know that the user study process has been completed, and all user-study results, along with their explanations and analysis, have been added to Table 2 in Section 4.2 of the main paper, and to Table 10 and Figure 16 in Appendix A.3. We now outline a summary of the details and results of the completed user study:

* **Participants:** We conducted a user-preference study with 115 participants. We distributed the questionnaire to two groups: experts and general participants. Of the 115 participants, 70 are general participants (including graduate students with computer science and engineering backgrounds), and 45 are experts: 31 software engineers, 3 project managers, 2 product owners, 5 UV/layout artists, and 4 modeling artists, all working in the Media and Entertainment industry for film and games.

* **Questionnaire:** Each participant completed 11 comparisons between textured 3D shapes and UV parameterizations produced by our method, FlexPara [1], OptCuts [2], Autodesk Maya [3], Blender [4], and xatlas [5]. For each comparison, participants rated each result according to three visual criteria: (i) texture-pattern smoothness/continuity, (ii) semantic alignment (how well colors correspond to meaningful parts), and (iii) seam visibility (how well seams are placed in occluded or less-obvious regions). The semantic-aware and visibility-aware evaluations consisted of 5 and 6 questions, respectively, for a total of 11 comparisons per participant.

* **Results:**

   * **Expert preference:** Expert participants, particularly UV, layout, and modeling artists, strongly prefer our proposed method over the baselines. Specifically, 42 out of 45 expert participants (93.70%) and 36 out of 45 expert participants (80.22%) prefer the results produced by our visibility-aware and semantic-aware methods, respectively.

   * **General preference:** General participants also strongly prefer our proposed method over the baselines. Specifically, 64 out of 70 general participants (91.42%) and 52 out of 70 general participants (74.29%) prefer the results produced by our visibility-aware and semantic-aware methods, respectively.

We have now added these explanations and results to the paper (updated on OpenReview), and all new content is highlighted in blue.

We thank reviewer dJSn again for suggesting the inclusion of a user study. It helped us evaluate our work with both industry-level professional users and general users, providing insight into how they perceive the results produced by our method. This has become another strong evaluation metric showing that our approach can produce UV parameterizations aligned with industry-level standards, and we hope our work serves as an important step toward developing more advanced tools for automatic UV parameterization and texture generation/editing.

---
### Local References:

[1] Zhao, Yuming, et al. "FlexPara: Flexible Neural Surface Parameterization." arXiv preprint arXiv:2504.19210 (2025).

[2] Li, Minchen, et al. "Optcuts: Joint optimization of surface cuts and parameterization." ACM transactions on graphics (TOG) 37.6 (2018): 1-13.

[3] Autodesk, Inc. Autodesk Maya (Version 2025.x). Autodesk, Inc., 2025. URL https://www.autodesk.com/products/maya/overview. Accessed: 2025-09-11

[4] Blender Foundation. Blender – a 3D creation suite (Version 4.5). Blender Foundation, 2025. URL https://www.blender.org. Accessed: 2025-09-11.

[5] jpcy. xatlas: Mesh parameterization / uv unwrapping library. https://github.com/jpcy/xatlas, 2025. Accessed: 2025-09-11.

---

### Meta-Review · Area_Chair_yqFm · 2026-01-05

**Summary:**

The paper initially received all positive ratings. The reviewers mentioned several weaknesses (see below). I believe most of them are well-addressed in the rebuttal. Thus I recommend acceptance.

**Reviewer Concerns:**

The main criticism are addressed in the rebuttal, such as lack of user study, comparison with commercial tools, incremental contribution,... The authors also admitted the method is not fully end-to-end differentiable.

**Reviewer Scores:**

Most concerns are addressed in the rebuttal. Based on my experience, reviewers would keep the current ratings or increased the scores a little bit.

---

### Decision · Program_Chairs · 2026-01-26

Accept (Poster)